# Poisson Midpoint Method for Log-Concave Sampling: Beyond the Strong Error Lower Bounds

**Rishikesh Srinivasan**
Google DeepMind
rishikeshsrini@google.com

**Dheeraj Nagaraj**
Google DeepMind
dheerajnagaraj@google.com

## ABSTRACT

We study the problem of sampling from strongly log-concave distributions over $\mathbb{R}^d$ using the Poisson midpoint discretization (a variant of the randomized midpoint method) for overdamped/underdamped Langevin dynamics. We prove its convergence in the 2-Wasserstein distance ($\mathcal{W}_2$), achieving a cubic speedup in dependence on the target accuracy ($\epsilon$) over the Euler-Maruyama discretization, surpassing existing bounds for randomized midpoint methods. Notably, in the case of underdamped Langevin dynamics, we demonstrate the complexity of $\mathcal{W}_2$ convergence is much smaller than the complexity lower bounds for convergence in $L^2$ strong error established in the literature.

## 1 INTRODUCTION

Sampling from a density $\pi(x) \propto \exp(-f(x))$ over $\mathbb{R}^d$ is of fundamental interest in physics, economics, and finance (Johannes & Polson, 2010; Von Toussaint, 2011; Kobyzev et al., 2020). Applications in computer science include volume computation (Vempala, 2010) and bandit optimization (Russo et al., 2018).

A popular approach is Langevin Monte Carlo (LMC) which is the Euler-Maruyama discretization of the continuous time Itô Stochastic Differential Equation (SDE) called (overdamped/underdamped) Langevin Dynamics. The convergence of LMC has been extensively studied in the literature (Durmus et al., 2019; Vempala & Wibisono, 2019; Erdogdu et al., 2022; Cheng & Bartlett, 2018; Cheng et al., 2018; Dalalyan & Riou-Durand, 2020; Altschuler et al., 2025) under various assumptions on the target density $\pi$, such as log-concavity and isoperimetry. The randomized midpoint discretization for Langevin dynamics (RLMC), introduced by Shen & Lee (2019) and developed further by Yu et al. (2024); He et al. (2021); Altschuler & Chewi (2024); Altschuler et al. (2025) considers a more sophisticated alternative to LMC. This is a randomized discretization which reduces the bias in the estimation of the Itô integral while introducing variance, leading to faster convergence bounds than for LMC. The Poisson Midpoint Method for Langevin dynamics (PLMC) was introduced by Kandasamy & Nagaraj (2024) as a variant of RLMC. This work considered the convergence of PLMC under general conditions (beyond strong log-concavity and isoperimetry) for the total variation distance via entropic central limit theorem style arguments.

The literature has focused on understanding the sharp limits to the computational complexity of sampling for various classes of algorithms, in terms of various problem parameters. In the case of strongly log-concave sampling, the work of Cao et al. (2021) established lower bounds for the strong $L^2$ error of randomized algorithms which discretize Underdamped Langevin Dynamics (ULD). Strong $L^2$ error is the $L^2$ distance between the continuous time Itô SDE solution at time $T$ and the sampling algorithm output whenever they are driven by *same* Brownian motion. This demonstrated that RLMC is an optimal discretization of ULD with respect to dimension and accuracy (up to log factors), in terms of the strong $L^2$ error. However, sampling algorithm guarantees generally consider 'weak' notions of distance such as total variation distance, Wasserstein distance, or the KL divergence between the law of algorithm output and the target. In particular, Wasserstein-2 distance bounds can consider the $L^2$ distance between algorithm output and the continuous time SDE driven by *different but arbitrarily coupled* Brownian motions.

In this work, we revisit the complexity of PLMC for strongly log-concave sampling in order to obtain better insights into the fundamental computational limits of sampling algorithms. We provide a sharp analysis via coupling arguments to obtain better convergence guarantees, which involves a tight bound on the $\mathcal{W}_2$ distance between a Gaussian random-variable and a perturbed Gaussian random-variable. This is adopted from Alex Zhai's proof of the Central Limit Theorem in $\mathcal{W}_2$ distance (Zhai, 2018), and leads to a substantial improvement in convergence guarantees.

## 1.1 OUR CONTRIBUTIONS

We consider the computational complexity of sampling from a log-concave target distribution $\pi(x) \propto \exp(-f(x))$ over $\mathbb{R}^d$, with $f$ well-conditioned (Assumption 1) with condition number $\kappa$ and strong convexity constant $\alpha$. Many classes of algorithms have been proposed and studied to this end. We study PLMC, which is a randomized algorithm for the discretization of Langevin Dynamics, with access only to $\nabla f(x)$ for arbitrary $x \in \mathbb{R}^d$. The computational complexity is measured in terms of number of evaluations of $\nabla f(x)$ (the oracle complexity).

**Limits of Sampling:** Recent works have aimed to understand the best possible computational complexity of sampling such that $\mathcal{W}_2^2(\text{output}, \pi) \leq \frac{\epsilon^2 d}{\alpha}$ in terms of $\epsilon, d$ and $\alpha$. Cao et al. (2021) show that randomized algorithms which discretize ULD require an oracle complexity of $\Omega(\epsilon^{-2/3})$ to converge in strong $L^2$ error; and RLMC achieves this rate up to logarithmic factors. It was thus widely believed in the literature that the rate of $\tilde{\mathcal{O}}(\epsilon^{-2/3})$, achieved by RLMC, might also be the optimal convergence rate in $\mathcal{W}_2$. The main contribution of our work is that we show it is possible to obtain $\tilde{\mathcal{O}}(\epsilon^{-1/3})$ complexity. Specifically, we show that:

**1.** Overdamped PLMC has an oracle complexity of $\tilde{\mathcal{O}}\left[\frac{\kappa^{4/3} + \kappa d^{1/3}}{\epsilon^{2/3}}\right]$ (Corollary 1).

**2.** Underdamped PLMC has an oracle complexity of $\tilde{\mathcal{O}}\left[\frac{\kappa^{7/6} d^{1/6}}{\epsilon^{1/3}} + \frac{\kappa^{\frac{11p+6}{8p+6}} d^{\frac{p}{4p+3}}}{\epsilon^{\frac{p+2}{4p+3}}}\right]$ (Corollary 2).

Here $p \in \mathbb{N}$ is arbitrary. For $p \geq 3$, this gives a complexity of $\tilde{\mathcal{O}}(\epsilon^{-1/3})$.

The best known convergence rate for overdamped LMC (in $\mathcal{W}_2$) is an oracle complexity of $\tilde{\mathcal{O}}(\epsilon^{-2})$ Durmus et al. (2019). The convergence guarantee of $\tilde{\mathcal{O}}(\epsilon^{-2/3})$ for overdamped PLMC is thus a cubic improvement in $\epsilon$ dependence. The best known convergence rate for underdamped LMC (in $\mathcal{W}_2$) is an oracle complexity of $\tilde{\mathcal{O}}(\epsilon^{-1})$. The convergence rate of $\tilde{\mathcal{O}}(\epsilon^{-1/3})$ for underdamped PLMC is again a cubic improvement. A detailed comparison of results is in Tables 1 and 2.

Concurrent work (Altschuler et al., 2025, Theorem 5.11) claims an oracle complexity of $\tilde{\mathcal{O}}(\kappa^{5/6} d^{5/3}/\epsilon^{2/3})$ to achieve $\text{KL}(\text{output}||\pi) \leq \epsilon^2$ for RLMC. This implies a complexity of $\tilde{\mathcal{O}}(\kappa^{5/6} d^{4/3}/\epsilon^{2/3})$ to achieve $\mathcal{W}_2^2 \leq \frac{\epsilon^2 d}{\alpha}$ via the $T_2$ inequality. This improves the dependence on $\kappa$ from $\kappa^{7/6}$ to $\kappa^{5/6}$ as compared to prior works, but with a worse dependence on $d$ and the same complexity in $\epsilon$.

**Comparison to Strong Error Lower Bounds:** The work of Cao et al. (2021) proves a lower bound for the discretization error of underdamped Langevin dynamics via randomized algorithms. In particular, given a probability space $\Omega$, $f$ satisfying Assumption 1 and a Brownian Motion $B_t(\omega) : \Omega \to \mathbb{R}^d$, consider the strong solution to equation 2 given by $X_T(\omega) = [U_T(\omega), V_T(\omega)]$ for some $T > 0$. The algorithm $A$ to approximate $U_T(\omega)$ has oracle access to $(\nabla f(x), \int_0^t e^{\theta s} dB_t(\omega))$ for any $x \in \mathbb{R}^d$, $t \in [0, T]$ and $s \in \{0, 2\}$ along with independent randomness $\tilde{\omega} \in \tilde{\Omega}$. The algorithm queries the oracle with $(x, t)$ of choice multiple times to produce an estimate $A(f, \omega, \tilde{\omega})$ for $U_T(\omega)$. This includes the case of Underdamped RLMC and Underdamped LMC. Their main result demonstrates that $\inf_{A \in \mathcal{A}_N} \sup_f E_{\omega, \tilde{\omega}} \|U_T(\omega) - A(f, \omega, \tilde{\omega})\|^2 \gtrsim C(T, L, \alpha) \frac{d}{N^3}$, where $\mathcal{A}_N$ is the set of all randomized algorithms as above with $N$ oracle queries. This error is the strong $L^2$ error since the algorithm and the SDE are driven by the same Brownian motion. This shows that algorithms of the above class need $N = \tilde{\Omega}_\kappa(\frac{1}{\epsilon^{2/3}})$ oracle queries to achieve strong $L^2$ error $\frac{\epsilon^2 d}{\alpha}$ and Underdamped RLMC achieves this optimal bound.

However, sampling algorithm guarantees consider 'weak errors' which are distances between $\text{Law}(U_T(\omega))$ and $\text{Law}(A(f, \omega, \tilde{\omega}))$. In particular, the Wasserstein-2 distance is the infimum of $L^2$

errors when $U_T$ is driven by $B_t(\omega)$ and $A(\cdot)$ queries $B'_t(\omega)$ over all couplings of *distinct* Brownian motions $B_t(\omega)$ and $B'_t(\omega)$. Our results show Poisson ULMC queries the oracle $\tilde{\mathcal{O}}_{\kappa,d}(\frac{1}{\epsilon^{1/3}})$ times in expectation to achieve $\mathcal{W}_2^2(\mathrm{Law}(A(f,\omega,\tilde{\omega})),\pi) \leq \frac{\epsilon^2 d}{\alpha}$, a quadratic improvement over RLMC.

We note that Kandasamy & Nagaraj (2024) obtained a complexity upper bound of $\tilde{\mathcal{O}}_{d,\kappa}(\frac{1}{\sqrt{\epsilon}})$ for Underdamped PLMC under LSI assumptions for achieving $\mathrm{TV} \leq \epsilon$. The literature on sampling algorithms compares bounds of the form $\mathcal{W}_2^2 \leq \frac{\epsilon^2}{\alpha}$ to bounds of the form $\mathrm{TV} \leq \epsilon$ (see Section 2.2). Under this comparison our bound improves over prior art. However, we note that TV and $\mathcal{W}_2^2$ bounds cannot be directly related rigorously.

## 2 NOTATION AND PROBLEM SETUP

Let $\|\cdot\|$ denote the standard Euclidean norm over $\mathbb{R}^d$ for some $d$ and $\mathbf{I}_d$ denote the $d \times d$ identity matrix. The notation $x = \mathcal{O}(y)$ and $x \lesssim y$ mean there exists a universal constant $C > 0$ such that $x \leq Cy$, and $\tilde{\mathcal{O}}(\cdot)$ hides logarithmic factors. The notation $\mathcal{O}_a(\cdot), \Omega_a(\cdot)$ mean the same as $\mathcal{O}(\cdot), \Omega(\cdot)$ except that they hide log factors. The number of evaluations of $\nabla f$ by the algorithm is referred to as 'oracle complexity'. We call the number of arithmetic operations (such as addition and multiplication) required on top of the oracle queries as 'arithmetic complexity'. PLMC can be implemented such that arithmetic complexity $= O(d \times$ oracle complexity$)$ as shown in the sequel. Thus, as is common in the literature, we only report the oracle complexity guarantees. Let $\mathrm{Law}(X)$ denote the law of the random variable $X$. Given two probability measures $\mu$ and $\nu$, we let $D_{\mathrm{KL}}(\mu||\nu)$ denote the KL divergence and $\mathrm{TV}(\mu,\nu)$ denote the total variation distance between them.

Given a sequence of probability measures $\mu_i$ over $\mathcal{X}_i$, for $i \in [n]$, a coupling is a probability measure $\Gamma$ over the product space $\prod_i \mathcal{X}_i$ such that the marginal over $\mathcal{X}_j$ is $\mu_j$. A sequence of random variables $(X_i \sim \mu_i)$ are coupled if they are defined over a common probability space, since their joint law is a coupling of $(\mu_i)_{i \in [n]}$. The Wasserstein-2 distance between $\mu$ and $\nu$ is given by

$$\mathcal{W}_2^2(\mu,\nu) := \inf_{\zeta \in \Gamma(\mu,\nu)} \int \|x-y\|^2 d\zeta(x,y),$$

where $\Gamma(\mu,\nu)$ denotes the set of couplings of $\mu$ and $\nu$. We make the following assumptions on $f$.

**Assumption 1.** The function $f : \mathbb{R}^d \to \mathbb{R}$ is $\alpha$ strongly convex and $L$ smooth for some $\alpha, L > 0$. That is, $f$ is twice continuously differentiable over $\mathbb{R}^d$ and for every $x, y \in \mathbb{R}^d$, we have: $f(y) - f(x) \geq \langle \nabla f(x), y-x \rangle + \frac{\alpha}{2} \|x-y\|^2$ and $\|\nabla f(x) - \nabla f(y)\| \leq L\|x-y\|$.

The target distribution, given by the density $\pi(x) \propto \exp(-f(x))$, is then called strongly log-concave. Our goal is to sample a random variable $X \sim \mu$ such that:[1]

$$\mathcal{W}_2^2(\mu,\pi) \leq \frac{\epsilon^2 d}{\alpha}. \tag{1}$$

We define the condition number $\kappa := \frac{L}{\alpha}$. Our notion of complexity is the number of gradient calls of $F$, in terms of the problem parameters $\kappa, d$ and $\epsilon$.

### 2.1 LANGEVIN MONTE CARLO

Suppose we wish to sample from $\pi \propto \exp(-f(x))$ in $\mathbb{R}^d$.

**Overdamped LMC (OLMC)** with step-size $\eta$ is the discrete time algorithm defined by the following iterates:

$$X_{t+1} = X_t - \eta \nabla f(X_t) + \sqrt{2\eta} Z_t,$$

where $Z_t \in \mathbb{R}^d$ is an independent standard Gaussian. This is the Euler-Maruyama discretization of Overdamped Langevin dynamics (OLD):

$$dX_t = -\nabla f(X_t)dt + \sqrt{2}dB_t,$$

whose stationary distribution is $\pi$. (Roberts & Tweedie, 1996)

---

[1] scaling $\frac{d}{\alpha}$ as considered in Shen & Lee (2019).

**Underdamped LMC (ULMC):** Let $U_t \in \mathbb{R}^d$ denote position, and $V_t \in \mathbb{R}^d$ denote momentum. ULMC with step-size $\eta$ is defined via the following recursion:

$$\begin{bmatrix} U_{t+1} \\ V_{t+1} \end{bmatrix} = A(\eta) \begin{bmatrix} U_t \\ V_t \end{bmatrix} - G(\eta) \begin{bmatrix} \nabla f(U_t) \\ 0 \end{bmatrix} + \Gamma(\eta) Z_t,$$

where $Z_t \in \mathbb{R}^{2d}$ is an independent standard Gaussian, and

$$A(\eta) = \begin{bmatrix} \mathbf{I}_d & \frac{1}{\gamma}(1 - e^{-\gamma\eta})\mathbf{I}_d \\ 0 & e^{-\gamma\eta}\mathbf{I}_d \end{bmatrix}, \; G(\eta) = \begin{bmatrix} \frac{1}{\gamma}(\eta - \frac{1}{\gamma}(1 - e^{-\gamma\eta}))\mathbf{I}_d & 0 \\ \frac{1}{\gamma}(1 - e^{-\gamma\eta})\mathbf{I}_d & 0 \end{bmatrix},$$

$$\Gamma(\eta)^2 := \begin{bmatrix} \frac{2}{\gamma}\left(\eta - \frac{2}{\gamma}(1 - e^{-\gamma\eta}) + \frac{1}{2\gamma}(1 - e^{-2\gamma\eta})\right)\mathbf{I}_d & \frac{1}{\gamma}(1 - 2e^{-\gamma\eta} + e^{-2\gamma\eta})\mathbf{I}_d \\ \frac{1}{\gamma}(1 - 2e^{-\gamma\eta} + e^{-2\gamma\eta})\mathbf{I}_d & (1 - e^{-2\gamma\eta})\mathbf{I}_d \end{bmatrix}.$$

This is the Euler-Maruyama discretization of the underdamped Langevin dynamics:

$$dU_t = V_t dt, \; dV_t = -\gamma V_t dt - \nabla f(U_t) dt + \sqrt{2\gamma} dB_t. \tag{2}$$

The stationary distribution of these dynamics is $\pi(U, V) \propto \exp(-f(U) - \frac{1}{2}||V||^2)$. (Eberle et al., 2019; Dalalyan & Riou-Durand, 2020)

## 2.2 PRIOR WORK

Recent works have focused on the rigorous theoretical analysis of classical and popular MCMC algorithms to establish complexity bounds and theoretical limits. The prototypical case of Overdamped LMC has been studied when the target $\pi$ is strongly log-concave and more generally when $\pi$ satisfies isoperimetric inequalities (Dalalyan, 2017; Durmus & Moulines, 2017; Durmus et al., 2019; Vempala & Wibisono, 2019; Erdogdu & Hosseinzadeh, 2021; Mou et al., 2022; Balasubramanian et al., 2022). Underdamped LMC has been considered as a faster alternative. This case too has been well studied when $\pi$ is strongly log-concave and when $\pi$ satisfies isoperimetric inequalities (Cheng et al., 2018; Dalalyan & Riou-Durand, 2020; Ganesh & Talwar, 2020; Ma et al., 2021; Zhang et al., 2023; Altschuler et al., 2025)

LMC is the Euler-Maruyama discretization of continuous time Langevin Dynamics, which can lead to sub-optimal convergence due to statistical bias in the approximation. Thus, Shen & Lee (2019) introduced the randomized midpoint method for LMC (RLMC) which reduces the bias in the approximation by introducing a randomized estimator at the cost of higher variance. RLMC does not involve higher order derivatives of $\nabla f$ as in Runge-Kutta schemes for SDEs (Kloeden et al., 1992) - allowing its use for generative modeling with denoising diffusion models (Kandasamy & Nagaraj, 2024). This leads to improvement in the convergence rates compared to LMC under log concavity (see Tables 1 and 2). He et al. (2021); Yu et al. (2024); Altschuler & Chewi (2024); Altschuler et al. (2025) extend the bounds in Shen & Lee (2019).

Kandasamy & Nagaraj (2024) introduced the Poisson midpoint method for LMC (PLMC), a variant of RLMC, which converges whenever LMC converges, allowing analysis beyond log-concavity. PLMC gives a quadratic improvement in complexity in terms of $\epsilon$ when $\pi$ satisfies Logarithmic Sobolev Inequalities (LSI). Our work shows a cubic improvement for PLMC under strong log-concavity.

The literature on MCMC considers various notions of convergence including KL-divergence, TV and $\mathcal{W}_2$. In the case when $\pi$ is strongly log-concave, the Otto-Villani Theorem (Otto & Villani, 2000) shows that $D_{\mathrm{KL}}(\mu||\pi) \leq \epsilon^2 \implies \mathcal{W}_2^2(\mu, \pi) \lesssim \frac{\epsilon^2}{\alpha}$ and the Pinsker's inequality shows that $D_{\mathrm{KL}}(\mu||\pi) \leq \epsilon^2 \implies \mathsf{TV}(\mu, \pi) \lesssim \epsilon$. The condition of $\pi$ satisfying LSI is more general than strong log-concavity of the target. We refer to Tables 1 and 2 for a detailed comparison of the results.

## 2.3 POISSON MIDPOINT METHOD

The Poisson midpoint method is a discrete variant of the randomized midpoint method introduced by Shen & Lee (2019). The iterates of PLMC run in batches of size $k$; and can be interpreted as a stochastic approximation of Langevin Monte-Carlo, with step-size $\eta/k$. Let $t$ and $i$ be integers, with $t \geq 0$ and $0 \leq i \leq k - 1$.

Table 1: Complexity for discretized OLD. In case of LSI, $\kappa = L \times$ LSI constant. The scaling of $\mathcal{W}_2^2$ is different from equation 1 in order to compare with TV and KL bounds.

| Overdamped Langevin Dynamics | | | |
|---|---|---|---|
| Algorithm | Assumption | Metric | Oracle Complexity |
| LMC Durmus et al. (2019) | Strongly Log-Concave | $\mathcal{W}_2^2 \leq \frac{\epsilon^2}{\alpha}$ | $\frac{\kappa d}{\epsilon^2}$ |
| RLMC Shen & Lee (2019); Yu et al. (2024) | Strongly Log-Concave | $\mathcal{W}_2^2 \leq \frac{\epsilon^2}{\alpha}$ | $\frac{\kappa \sqrt{d}}{\epsilon} + \frac{\kappa^{4/3} d^{1/3}}{\epsilon^{2/3}}$ |
| RLMC Altschuler & Chewi (2024) | Strongly Log-Concave | KL $\leq \epsilon^2$ | $\frac{\kappa \sqrt{d}}{\epsilon}$ |
| RLMC Altschuler & Chewi (2024) | LSI | KL $\leq \epsilon^2$ | $\frac{\kappa^{3/2} \sqrt{d}}{\epsilon}$ |
| PLMC (Ours) | Strongly Log-Concave | $\mathcal{W}_2^2 \leq \frac{\epsilon^2}{\alpha}$ | $\frac{\kappa^{4/3} d^{1/3} + \kappa d^{2/3}}{\epsilon^{2/3}}$ |

Table 2: Complexity for discretized ULD. In case of LSI, $\kappa = L \times$ LSI constant. The scaling of $\mathcal{W}_2^2$ is different from equation 1 in order to compare with TV and KL bounds, and $p \in \mathbb{N}$ is arbitrary.

| Underdamped Langevin Dynamics | | | |
|---|---|---|---|
| Algorithm | Assumption | Metric | Oracle Complexity |
| LMC Dalalyan & Riou-Durand (2020) | Strongly Log-Concave | $\mathcal{W}_2^2 \leq \frac{\epsilon^2}{\alpha}$ | $\frac{\kappa^{3/2} \sqrt{d}}{\epsilon}$ |
| RLMC Shen & Lee (2019); Yu et al. (2024) | Strongly Log-Concave | $\mathcal{W}_2^2 \leq \frac{\epsilon^2}{\alpha}$ | $\frac{\kappa d^{1/3}}{\epsilon^{2/3}} + \frac{\kappa^{7/6} d^{1/6}}{\epsilon^{1/3}}$ |
| PLMC Kandasamy & Nagaraj (2024) | LSI | TV $\leq \epsilon$ | $\frac{\kappa^{\frac{17}{12}} d^{\frac{5}{12}}}{\sqrt{\epsilon}}$ |
| PLMC (Ours) | Strongly Log-Concave | $\mathcal{W}_2^2 \leq \frac{\epsilon^2}{\alpha}$ | $\frac{\kappa^{7/6} d^{1/3}}{\epsilon^{1/3}} + \frac{\kappa^{\frac{11p+6}{8p+6}} d^{\frac{3p+2}{8p+6}}}{\epsilon^{\frac{p+2}{4p+3}}}$ |

To emphasize the comparison with PLMC, we adopt the following notation for **overdamped LMC:**

$$X_{t,i+1} = X_{t,i} - \frac{\eta}{k} \nabla f(X_{t,i}) + \sqrt{\frac{2\eta}{k}} Y_{t,i},$$

$$X_{t+1,0} = X_{t,k}.$$

Here $Y_{t,i} \in \mathbb{R}^d$ denote independent standard Gaussians. Note that this is OLMC with step-size $\eta/k$, grouped into batches of size $k$. Now let $Z_{t,i} \in \mathbb{R}^d$ be independent standard Gaussians, and $H_{t,i}$ be independent Bernoulli random variables with parameter $1/k$.

**Overdamped PLMC** is defined by the following recursions:

$$\tilde{X}_{t,i}^+ = \tilde{X}_{t,0} - \frac{\eta i}{k} \nabla f(\tilde{X}_{t,0}) + \sum_{j=0}^{i-1} \sqrt{\frac{2\eta}{k}} Z_{t,j}$$

$$\tilde{X}_{t,i+1} = \tilde{X}_{t,i} - \frac{\eta}{k} \nabla f(\tilde{X}_{t,0}) + \eta H_{t,i}(\nabla f(\tilde{X}_{t,0}) - \nabla f(\tilde{X}_{t,i}^+)) + \sqrt{\frac{2\eta}{k}} Z_{t,i}$$

$$\tilde{X}_{t+1,0} = \tilde{X}_{t,k}$$

**Remark 1.** The iterates $\tilde{X}_{t,i}^+$ denote midpoints. They are defined the same way as in Shen & Lee (2019). The correction term $\eta H_{t,i}(\nabla f(\tilde{X}_{t,0}) - \nabla f(\tilde{X}_{t,i}^+))$ decides whether we use the gradient evaluated at our midpoint. In expectation over $H_{t,i}$, the drift term is $\frac{\eta}{k} \nabla f(\tilde{X}_{t,i}^+)$. However, we only need to evaluate $\nabla f(\tilde{X}_{t,i}^+)$ when $H_{t,i} = 1$. With $N_t = \sum_{i=0}^{k-1} H_{t,i}$ we have $\mathbb{E} N_t = 1$. This means we need an expected number of 2 gradient calls to $f$ per batch including $\nabla f(\tilde{X}_{t,0})$. This facilitates an efficient implementation of PLMC where $\tilde{X}_{t+1,0}$ can be computed directly from $\tilde{X}_{t,0}$, with $\tilde{\mathcal{O}}(1)$ gradient calls and an arithmetic complexity of $\tilde{\mathcal{O}}(d)$. This relies on the properties of jointly Gaussian

random variables, and is detailed in Kandasamy & Nagaraj (2024, Section 2.2). This is explicated to the case of overdamped PLMC in Algorithm 1. [2]

---

**Algorithm 1** Efficient Implementation of Overdamped PLMC Step.

---

**Step 1.** Generate $\mathbb{I}_t = \{i_1, \ldots, i_{N_t}\}$ such that $H_{t,i} = 1$ if and only if $i \in \mathbb{I}_t$, and $i_1 < \cdots < i_{N_t}$

**Step 2.** $m_{t,0} \leftarrow 0$, $Z_{t,n} \sim \mathcal{N}(0, \mathbf{I})$ i.i.d. $i_0 \leftarrow 0$, $i_{N_t+1} \leftarrow k-1$. For $1 \le n \le N_t + 1$:

$$m_{t,n} \leftarrow m_{t,n-1} + \sqrt{\frac{2\eta(i_n - i_{n-1})}{k}} Z_{t,n},$$

**Step 3.** For $1 \le n \le N_t$,

$$\tilde{X}_{t,i_n}^+ \leftarrow \tilde{X}_{t,0} - \frac{\eta i_n}{k}\nabla f(\tilde{X}_{t,0}) + m_{t,n}.$$

**Step 4.**

$$\Delta_t \leftarrow \frac{\eta}{k}\sum_{n=1}^{N_t}(\nabla f(\tilde{X}_{t,0}) - \nabla f(\tilde{X}_{t,i_n}^+))$$

**Step 5.**

$$\tilde{X}_{t+1,0} \leftarrow \tilde{X}_{t,0} - \frac{\eta}{k}\nabla f(\tilde{X}_{t,0}) + \Delta_t + m_{t,N_t+1}$$

---

**Underdamped PLMC** is defined in a similar manner, by the following recursions:

$$\begin{bmatrix}\tilde{U}_{t,i}^+\\\tilde{V}_{t,i}^+\end{bmatrix} = A\Big(\frac{\eta i}{k}\Big)\begin{bmatrix}\tilde{U}_{t,0}\\\tilde{V}_{t,0}\end{bmatrix} - G\Big(\frac{\eta i}{k}\Big)\begin{bmatrix}\nabla f(U_{t,0})\\0\end{bmatrix} + \sum_{j=0}^{i-1}A\Big(\frac{\eta(i-1-j)}{k}\Big)\Gamma\Big(\frac{\eta i}{k}\Big)Z_{t,i}$$

$$\begin{bmatrix}\tilde{U}_{t,i+1}\\\tilde{V}_{t,i+1}\end{bmatrix} = A\Big(\frac{\eta}{k}\Big)\begin{bmatrix}\tilde{U}_{t,i}\\\tilde{V}_{t,i}\end{bmatrix} - G\Big(\frac{\eta}{k}\Big)\begin{bmatrix}\nabla f(\tilde{U}_{t,0})\\0\end{bmatrix} + \Gamma\Big(\frac{\eta}{k}\Big)Z_{t,i} + kH_{t,i}\cdot G\Big(\frac{\eta}{k}\Big)\begin{bmatrix}\nabla f(\tilde{U}_{t,0}) - \nabla f(\tilde{U}_{t,i}^+)\\0\end{bmatrix}$$

$$\begin{bmatrix}\tilde{U}_{t+1,0}\\\tilde{V}_{t+1,0}\end{bmatrix} = \begin{bmatrix}\tilde{U}_{t,k}\\\tilde{V}_{t,k}\end{bmatrix}$$

With $A$, $G$ and $\Gamma$ as defined in 2.1, and $Z_{t,i} \in \mathbb{R}^d$ being independent standard Gaussians.

**Remark 2.** As in the overdamped case, $\tilde{U}_{t,i}^+$ and $\tilde{V}_{t,i}^+$ denote midpoints, and the outcome of the Bernoulli decides whether we evaluate the gradient at the midpoint. We note that the comments on complexity in Remark 1 are also valid in the underdamped case. An efficient implementation of underdamped PLMC is given in Algorithm 2.

We adopt the following notation for **underdamped LMC**, to emphasize the comparison to PLMC.

$$\begin{bmatrix}U_{t,i+1}\\V_{t,i+1}\end{bmatrix} = A\Big(\frac{\eta}{k}\Big)\begin{bmatrix}U_{t,i}\\V_{t,i}\end{bmatrix} - G\Big(\frac{\eta}{k}\Big)\begin{bmatrix}\nabla f(U_{t,i})\\0\end{bmatrix} + \Gamma\Big(\frac{\eta}{k}\Big)Y_{t,i},$$

$$\begin{bmatrix}U_{t+1,0}\\V_{t+1,0}\end{bmatrix} = \begin{bmatrix}U_{t,k}\\V_{t,k}\end{bmatrix},$$

where $Y_{t,i} \in \mathbb{R}^{2d}$ is an independent standard Gaussian. Note that this is underdamped LMC with step-size $\eta/k$, grouped into batches of size $k$.

## 3 RESULTS

We now present our main results. The following Theorem on the convergence of overdamped PLMC is proven in Section C.

**Theorem 1.** Let $\tilde{X}_{t,i}$ denote the iterates of Overdamped PLMC, and $X_{t,i}$ the iterates of Overdamped LMC with stepsize $\eta/k$, as defined in Section 2.3. Assume $\eta L \le 1/8$, and Assumption 1. Then there exist absolute constants $c_1$ and $c_2$ such that

$$\mathcal{W}_2^2(\mathrm{Law}(\tilde{X}_{t,0}), \mathrm{Law}(X_{t,0})) \lesssim (\eta^6 L^4 dk + \eta^4 L^2 + \tfrac{\eta^5 L^4}{\alpha})\cdot(Ldt + \tfrac{1}{\eta}\mathbb{E}(f(X_{0,0}) - f(X_{t,0}))$$

$$+ \tfrac{\eta^3 L^4 d}{\alpha^2} + \tfrac{\eta^4 L^4 d^2}{\alpha} + \exp(c_1 d - (c_2\eta^2 L^2 k)^{-1})\cdot\tfrac{\eta^2 L^2 d}{\alpha}.$$

---

[2]The original paper contains a typo, which has been rectified in our exposition.

The above theorem shows that $\tilde{X}_{t,0}$ is close to $X_{t,0}$ in Wasserstein-2 distance. However, running $tk$ iterations of PLMC requires only $\mathcal{O}(t)$ gradient calls, as compared to $tk$ gradient calls for LMC. In the following corollary, we combine the Theorem 1 with the convergence results for $X_{t,i}$ to $\pi$ (given in Durmus et al. (2019)) to deduce the convergence of $\tilde{X}_{t,i}$. We refer to Section D.4 for its proof.

**Corollary 1.** Let $\tilde{X}_{t,0}$ be the iterates of Overdamped PLMC as in Theorem 1. Let $x^*$ be the unique minimizer of $f$, and $\epsilon > 0$. Assume:

1. The conditions from Theorem 1 hold.

2. $\tilde{X}_{0,0}$ satisfies $\mathbb{E}[f(\tilde{X}_{0,0}) - f(x^*)] \leq C_f \kappa d$ for some $C_f > 0$.

Then there exist constants $C_1, C_2 > 0$ depending only on $C_f$, $\log(\frac{W_2(X_{0,0},\pi)\sqrt{\alpha}}{\epsilon\sqrt{d}})$ and $\log(1/\epsilon)$, polynomially, such that if $\eta = C_1 \min(\frac{\alpha^{1/3}\epsilon^{2/3}}{L^{4/3}}, \frac{\epsilon^{2/3}}{d^{1/3}L})$, $k \asymp \max(\frac{\eta L}{\epsilon^2}, 1)$ and $N = C_2 \left[\frac{\kappa^{4/3}+\kappa d^{1/3}}{\epsilon^{2/3}}\right]$. Then,
$$\mathcal{W}_2^2(\mathrm{Law}(\tilde{X}_{N,0}), \pi) \leq \epsilon^2 d/\alpha$$

**Remark 3.** The complexity bound for Overdamped LMC (Durmus et al., 2019) is $\tilde{\mathcal{O}}(\kappa/\epsilon^2)$ gradient calls, and that of Overdamped RLMC (Yu et al., 2024) is $\tilde{\mathcal{O}}(\frac{\kappa}{\epsilon} + \frac{\kappa^{4/3}}{\epsilon^{2/3}})$ gradient calls. To our knowledge, our method is thus the best known discretization of overdamped Langevin dynamics, in terms of $\epsilon$ dependence. Note that our assumption on the initialization is very mild - $f$ can be optimized easily using standard convex optimization algorithms.

The following Theorem, proved in Section F, considers Underdamped Langevin Dynamics:

**Theorem 2.** Let $\tilde{U}_{t,i}$ denote the iterates of Underdamped PLMC, and $U_{t,i}$ denote the iterates of Underdamped LMC with step-size $\eta/k$, as defined in Section 2.3. Let $p \geq 0$ be any integer. There exists $c_0 > 0$, which depends only on $p$, such that if:

1. Assumption 1 holds.

2. $\gamma\eta < c_0$, $\frac{\eta}{k} \leq \frac{c_0}{\kappa\sqrt{L}}$, and $\frac{\eta^{3p-1}t^{p-1}L^{2p}}{\gamma^{p+1}} < c_0$

3. $\gamma = c_\gamma\sqrt{L}$ for some constant $c_\gamma \geq \sqrt{2}$.

Then, $\quad \mathcal{W}_2^2(\mathrm{Law}(\tilde{U}_{t,i}), \mathrm{Law}(U_{t,i})) = \mathcal{O}\left[\frac{\eta^7 L^{9/2}d}{\alpha\gamma^2}t + \frac{\eta^8 L^4 d^2}{\gamma^2}t^2 + \frac{\eta^{4p+4}k^{p-1}L^{2p+2}d^{p+1}}{\gamma^2}t^{p+1}\right]$
$$+ \mathbb{E}[P_\eta(||V_{0,0}||, |f(\Psi_0) - f(\Psi_t)|^+)],$$

Where $\mathcal{O}$ hides constants depending only on $c_0, c_\gamma$. $P_\eta$ is a polynomial whose coefficients are high powers of $\eta$ and depend on $p, c_\gamma$, and $\Psi$ is defined as $\Psi_s := \tilde{U}_{s,0} + \frac{1}{\gamma}\tilde{V}_{s,0}$. The complete bound is explicated in Section F.3, for the sake of clarity.

The bound in Theorem 2 holds for any choice of nonnegative integer $p$. The presence of $p$ is due to the manner in which we bound a certain low probability event - see the proof of Proposition 3. Similar to Corollary 1, the following Corollary (proved in Section H) establishes complexity bounds.

**Corollary 2.** Let $\tilde{U}_{t,i}$ denote the iterates of Underdamped PLMC, as in Theorem 2. Let $x^*$ be the unique minimizer of $f$, and $p \in \mathbb{N} \cup \{0\}$ be fixed. Let $k \asymp \max(\lceil\frac{\eta L}{\epsilon\sqrt{\alpha}}\rceil, 1)$, and $\gamma = c_\gamma\sqrt{L}$ as in Theorem 2. Initialize the iterates with $V_{0,0} \sim \mathcal{N}(0, \mathbf{I}_d)$ and $\mathbb{E}||U_{0,0} - x^*||^{2n} \leq c_1 d^n/L^n$ for $n = \max(2, p+1)$, and some constant $c_1 > 0$ depending only on $p$.

Then there exist $C_3, C_4, C_5 > 0$ depending on $p$ and polynomially on $\log(\frac{\mathcal{W}_2(U_0,\pi)\sqrt{\alpha}}{\epsilon\sqrt{d}})$ and $\log(\frac{\kappa}{\epsilon})$, such that: if $\eta = C_3 \min\left(\frac{\epsilon^{1/3}}{\kappa^{1/6}d^{1/6}\sqrt{L}}, \frac{\epsilon^{\frac{p+2}{4p+3}}}{\kappa^{\frac{3p}{8p+6}}d^{\frac{p}{4p+3}}\sqrt{L}}\right)$, $0 < \epsilon \leq C_4 \min(\kappa^{-1/2}, \kappa^{-\frac{2p-3}{2p}}d^{1/2})$ and $N = C_5\left[\frac{\kappa^{7/6}d^{1/6}}{\epsilon^{1/3}} + \frac{\kappa^{\frac{11p+6}{8p+6}}d^{\frac{p}{4p+3}}}{\epsilon^{\frac{p+2}{4p+3}}}\right]$, we have
$$\mathcal{W}_2^2(\mathrm{Law}(\tilde{U}_{N,0}), \pi) \leq \epsilon^2 d/\alpha\,.$$

The complexity bound for Underdamped LMC is $\tilde{\mathcal{O}}(\kappa^{3/2}/\epsilon)$ (Cheng et al., 2018), and that of Underdamped RLMC is $\tilde{\mathcal{O}}(\frac{\kappa^{7/6}}{\epsilon^{1/3}} + \frac{\kappa}{\epsilon^{2/3}})$ (Shen & Lee, 2019).

**Remark 4.** Our assumption on the initialization is standard in the literature (Vempala & Wibisono, 2019; Shen & Lee, 2019), and satisfied (for example) by $\mathcal{N}(x^*, \mathbf{I}_d/L)$.

1. With $p = 0$, we get a complexity of $\tilde{\mathcal{O}}(\frac{\kappa^{7/6}d^{1/6}}{\epsilon^{1/3}} + \frac{\kappa}{\epsilon^{2/3}})$.

2. With $p = 3$, we get a complexity of $\tilde{\mathcal{O}}(\frac{\kappa^{13/10}d^{1/5}}{\epsilon^{1/3}})$.

3. For $p > 3$, the second term becomes lower order in $\epsilon$ and the oracle complexity satisfies $\tilde{\mathcal{O}}_{\kappa,d}(\frac{1}{\epsilon^{1/3}} + \frac{1}{\epsilon^{1/4+\mathcal{O}(1/p)}})$.

**Remark 5.** The concurrent work of Altschuler et al. (2025) claims an oracle complexity of $\tilde{\mathcal{O}}(\kappa^{5/6}d^{5/3}/\epsilon^{2/3})$ to achieve KL $\leq \epsilon^2$. This is in the low friction regime $\gamma \asymp \sqrt{\alpha}$, and for a double midpoint implementation of Underdamped RLMC. This has improved dependence in $\kappa$ as compared to prior works, but is worse in $d$ and without improvement in $\epsilon$.

Our work improves dependence in $\epsilon$ while being worse in $d$. We believe the dependence on $\kappa$ can be improved if we obtain tight bounds on the higher order moments of our algorithm (Remark 8). To our knowledge, PLMC is the best known discretization of underdamped Langevin dynamics in terms of $\epsilon$, and is the first known algorithm to break the $\tilde{\mathcal{O}}(\epsilon^{-2/3})$ barrier for strongly log-concave sampling.

## 4 INTUITION AND PROOF IDEA

Our proof relies on the following key Lemma. This is similar to Lemma 7 of Kandasamy & Nagaraj (2024), which was in turn adapted from Zhai (2018). The difference is that our result avoids higher order moments, making it significantly easier to apply.

**Lemma 1.** Let $V$ be a random vector in $\mathbb{R}^d$ satisfying the following conditions:

1. $||V|| \leq \beta$ a.s., $\mathbb{E}[V] = 0$, and $\mathbb{E}[VV^T] = \Sigma$.

2. $V$ lies in a one-dimensional subspace almost surely.

Let the random vector $Z \sim \mathcal{N}(0, \mathbf{I}_d)$, and independent of $V$. Let $\nu = \text{Tr}(\Sigma)$, Then,

$$\mathcal{W}_2^2(\text{Law}(Z), \text{Law}(Z+V)) \leq \frac{11}{2}\nu^2 + \mathbf{1}_{5\beta^2 > 1} \cdot 2\nu.$$

A naive bound would be $\mathcal{W}_2^2(\text{Law}(Z), \text{Law}(Z+V)) \leq \nu$, which corresponds to the Gaussians being coupled identically. Note that $\nu^2$ can be much smaller than $\nu$, and this leads to our sharp result. To intuitively explain this bound, we consider the scenario where $V \sim \mathcal{N}(0, \nu) \in \mathbb{R}$, and $Z \sim \mathcal{N}(0, 1)$. In this case, Lemma 1 does not apply, but there is a closed form expression for $\mathcal{W}_2(\text{Law}(Z), \text{Law}(Z+V))$.

$$\mathcal{W}_2^2(\text{Law}(Z), \text{Law}(Z+V)) = 2 + \nu - 2\sqrt{1+\nu} = \Theta(\nu^2), \quad \text{when } \nu << 1.$$

**Interpreting overdamped PLMC as LMC with perturbed Gaussian noise.** From the definition in Section 2.3, overdamped PLMC can be written as follows.

$$\tilde{X}_{t,i+1} = \tilde{X}_{t,i} - \frac{\eta}{k}\nabla f(\tilde{X}_{t,i}) + \sqrt{\frac{2\eta}{k}}\tilde{Z}_{t,i},$$

where $\tilde{Z}_{t,i}$ denotes the perturbed Gaussian and is given by the following expression.

$$\tilde{Z}_{t,i} = \sqrt{\frac{\eta k}{2}}(H_{t,i} - 1/k)(\nabla f(\tilde{X}_{t,0}) - \nabla f(\tilde{X}_{t,i}^+)) + \sqrt{\frac{\eta}{2k}}(\nabla f(\tilde{X}_{t,i}) - \nabla f(\tilde{X}_{t,i}^+)) + Z_{t,i}.$$

Conditioned on the previous iterates $\tilde{X}_{t,0}, \tilde{X}_{t,i}^+$ and $\tilde{X}_{t,i}$, this is a Gaussian with mean $B_{t,i} = \sqrt{\frac{\eta}{2k}}(\nabla f(\tilde{X}_{t,i}) - \nabla f(\tilde{X}_{t,i}^+))$, perturbed by the zero-mean random vector $S_{t,i} = \sqrt{\frac{\eta k}{2}}(H_{t,i} - $

$1/k)(\nabla f(\tilde{X}_{t,0}) - \nabla f(\tilde{X}_{t,i}^+))$. Note that $S_{t,i}$ lies in a one dimensional subspace a.s., since it is determined by the Bernoulli $(H_{t,i} - 1/k)$.

**Gradient descent is contractive.** Given $\eta < 1$, and that $f$ is well-conditioned (Assumption 1), the map $T(x) = x - \eta \nabla f(x)$ is Lipschitz with parameter $(1 - \alpha \eta)$.

**Constructing a coupling.** As seen in Section 2.3, iterates of Langevin Monte-Carlo are defined by

$$X_{t,i+1} = X_{t,i} - \frac{\eta}{k} \nabla f(X_{t,i}) + \sqrt{\frac{2\eta}{k}} Y_{t,i}.$$

In order to couple $X_{t,i+1}$ and $\tilde{X}_{t,i+1}$, we first let $X_{t,i}$ and $\tilde{X}_{t,i}$ be coupled optimally. Conditioned on $X_{t,i}, \tilde{X}_{t,i}, \tilde{X}_{t,i}^+$ and $\tilde{X}_{t,0}$, we couple $Y_{t,i}$ and $\tilde{Z}_{t,i}$ optimally as per the bound established in Lemma 1. This allows us to produce a recursion of the following form.

$$\mathcal{W}_2^2(\text{Law}(X_{t,i+1}), \text{Law}(\tilde{X}_{t,i+1})) \le (1 - \frac{\alpha \eta}{2k}) \mathcal{W}_2^2(\text{Law}(X_{t,i}), \text{Law}(\tilde{X}_{t,i})) + E_{t,i},$$

where $E_{t,i}$ is an appropriate bound on the discretization error.

**Bounding the discretization error.** The application of the CLT as detailed above gives us terms of the form $\mathbb{E}||\tilde{X}_{t,i} - \tilde{X}_{t,0}||^p$ and $\mathbb{E}||\tilde{X}_{t,i}^+ - \tilde{X}_{t,0}||^p$ for some $p \in \mathbb{N}$. These can be bounded in terms of $\mathbb{E}||\nabla f(\tilde{X}_{t,0})||^p$ and Gaussian moments. We then reduce the bounds to $\mathbb{E}||\nabla f(\tilde{X}_{t,0})||^2$ rather than $\mathbb{E}||\nabla f(\tilde{X}_{t,0})||^p$, and then apply the following gradient bound, which we believe is tight.

**Lemma 2.** Assuming $\eta L \le 1/8$, the following bound is true.

$$\sum_{t=0}^{N-1} \mathbb{E}||\nabla f(\tilde{X}_{t,0})||^2 \lesssim \frac{1}{\eta} \mathbb{E}[f(\tilde{X}_{0,0}) - f(\tilde{X}_{N,0})] + LdN.$$

This is proven in Section D.3. It is known (Vempala & Wibisono, 2019, Lemma 11) that $\int_{\mathbb{R}^d} ||\nabla f(x)||^2 d\pi(x) \le Ld$ under smoothness. This bound is tight when $\pi$ is Gaussian. Therefore, we expect that the dominant term $LdN$ in our bound cannot be improved at stationarity.

**The underdamped case.** We make the following coordinate change for the iterates of underdamped LMC/PLMC.

$$\begin{bmatrix} x \\ y \end{bmatrix} \to \mathcal{M} \begin{bmatrix} x \\ y \end{bmatrix}, \text{ where } \mathcal{M} = \begin{bmatrix} \mathbf{I}_d & 0 \\ \mathbf{I}_d & \frac{2}{\gamma} \mathbf{I}_d \end{bmatrix}.$$

Under this transformation, and with appropriate step-size; the deterministic component of the ULMC recursion is contractive. For a precise statement, see Lemma 16 of Zhang et al. (2023). We denote $W_{t,i} = U_{t,i} + \frac{2}{\gamma} V_{t,i}$, and $X_{t,i} = [U_{t,i}, W_{t,i}]^T$.

Under our transformation $\mathcal{M}$, for appropriate matrices $A_{\mathcal{M}}, G_{\mathcal{M}}, \Gamma_{\mathcal{M}}$ defined in Section F, we have:

$$X_{t,i+1} = A_{\mathcal{M}}\left(\frac{\eta}{k}\right) \begin{bmatrix} U_{t,i} \\ W_{t,i} \end{bmatrix} - G_{\mathcal{M}}\left(\frac{\eta}{k}\right) \begin{bmatrix} \nabla f(U_{t,i}) \\ 0 \end{bmatrix} + \Gamma_{\mathcal{M}}\left(\frac{\eta}{k}\right) Y_{t,i},$$

$$X_{t+1,0} = X_{t,k}$$

This allows the ULMC recursion to be interpreted as a noisy contraction similar to OLMC. Define $T : \mathbb{R}^{2d} \to \mathbb{R}^{2d}$ by

$$T \begin{bmatrix} u \\ w \end{bmatrix} = A_{\mathcal{M}}(\eta) \begin{bmatrix} u \\ w \end{bmatrix} - G_{\mathcal{M}}(\eta) \begin{bmatrix} \nabla f(u) \\ 0 \end{bmatrix}.$$

Then $T$ is Lipschitz with constant $(1 - \frac{\alpha \eta}{\gamma} + L\eta^2)$ (Zhang et al., 2023, Lemma 16), and is hence contractive for small $\eta$. Using this perspective, we are able to follow a similar proof technique as in the overdamped case. In this case, we require bounds on the moments $\mathbb{E}||\nabla f(\tilde{U}_{t,0})||^p$ and $\mathbb{E}||\tilde{V}_{t,0}||^p$. We use Theorem 4, to bound these moments.

## 5 CONCLUSION:

We considered the Poisson Midpoint discretization of Overdamped and Underdamped Langevin Dynamics, and showed state of the art oracle complexity of $\tilde{\mathcal{O}}_{\kappa,d}\left(\frac{1}{\epsilon^{1/3}}\right)$ for convergence in the Wasserstein-2 distance to the strong log-concave stationary law $\pi$. This breaks the conjectured lower bound of $\tilde{\Omega}_{\kappa,d}\left(\frac{1}{\epsilon^{2/3}}\right)$. Our work is an effort towards understanding the fundamental computational complexity of sampling from strongly log-concave distributions in terms of $\kappa, \epsilon$ and $d$, and shows an improved bound in terms of $\epsilon$. Concurrent work (Altschuler et al., 2025) claims an improvement of the state of the art dependence on $\kappa$ (from $\kappa^{7/6} \to \kappa^{5/6}$) but with a worse dependence on $\epsilon, d$. In future, we hope to explore techniques which simultaneously improve dependence on all three parameters. In particular, we believe our result can be improved in $\kappa$ if we obtain tight bounds on the moments $\mathbb{E}||\nabla f(\tilde{U}_{t,0})||^p$ and $\mathbb{E}||\tilde{V}_{t,0}||^p$ (Remark 8), and this is an avenue for future research.

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

## A   EFFICIENT IMPLEMENTATION OF UNDERDAMPED PLMC

---

**Algorithm 2** Efficient Implementation of Underdamped PLMC Step.

---

**Step 1.** Generate $\mathbb{I}_t = \{i_1, \ldots, i_{N_t}\}$ such that $H_{t,i} = 1$ iff $i \in \mathbb{I}_t$, and $i_1 < \cdots < i_{N_t}$ without loss of generality.

**Step 2.** Let $m_{t,0} \leftarrow 0$, and $Z_{t,n} \in \mathbb{R}^{2d}$ be a sequence of i.i.d. standard Gaussians. For $1 \leq n \leq N_t + 1$:

$$m_{t,n} \leftarrow A\Big(\frac{\eta(i_n - i_{n-1})}{k}\Big)m_{t,n-1} + \Gamma\Big(\frac{\eta(i_n - i_{n-1})}{k}\Big)Z_{t,n},$$

with the convention that $i_0 = 0$ and $i_{N_t+1} = k - 1$.

**Step 3.** For $1 \leq n \leq N_t$, compute

$$\begin{bmatrix} \tilde{U}_{t,i_n}^+ \\ \tilde{V}_{t,i_n}^+ \end{bmatrix} \leftarrow A\Big(\frac{\eta i_n}{k}\Big) \begin{bmatrix} \tilde{U}_{t,0} \\ \tilde{V}_{t,0} \end{bmatrix} - G\Big(\frac{\eta i_n}{k}\Big) \begin{bmatrix} \nabla f(\tilde{U}_{t,0}) \\ 0 \end{bmatrix} + m_{t,n}.$$

**Step 4.** Compute the correction term:

$$\Delta_t \leftarrow k \sum_{n=1}^{N_t} A\Big(\frac{\eta(k - 1 - i_n)}{k}\Big) G\Big(\frac{\eta}{k}\Big) \begin{bmatrix} \nabla f(\tilde{U}_{t,0}) - \nabla f(\tilde{U}_{t,i_n}^+) \\ 0 \end{bmatrix}$$

**Step 5.** Compute $\tilde{U}_{t+1,0}$ and $\tilde{V}_{t+1,0}$ :

$$\begin{bmatrix} \tilde{U}_{t+1,0} \\ \tilde{V}_{t+1,0} \end{bmatrix} \leftarrow A(\eta) \begin{bmatrix} \tilde{U}_{t,0} \\ \tilde{V}_{t,0} \end{bmatrix} - G(\eta) \begin{bmatrix} \nabla f(\tilde{U}_{t,0}) \\ 0 \end{bmatrix} + \Delta_t + m_{t,N_t+1}.$$

---

## B   PROOF OF LEMMA 1

By the triangle inequality for $\mathcal{W}_2$, we have

$$\mathcal{W}_2^2(\mathrm{Law}(Z), \mathrm{Law}(Z + V)) \leq 2\mathcal{W}_2^2(\mathrm{Law}(Z), \mathrm{Law}(\sqrt{\mathbf{I}_d + \Sigma}Z))$$
$$+ 2\mathcal{W}_2^2(\mathrm{Law}(\sqrt{\mathbf{I}_d + \Sigma}Z), \mathrm{Law}(Z + V))$$

The latter term is a Wasserstein distance between Gaussians, which has the following closed form.

$$2\mathcal{W}_2^2(\mathrm{Law}(\sqrt{\mathbf{I}_d + \Sigma}Z), \mathrm{Law}(Z)) = 4 + 2\nu - 4\sqrt{1 + \nu} \leq \frac{1}{2}\nu^2.$$

The former term is bounded below (Lemma 4), using a key result due to Alex Zhai. We check that the proof by Zhai (2018, Lemma 1.6) does not require $n$ to be an integer and state the following:

**Lemma 3** (Lemma 1.6, Zhai (2018)). Let $n > 0$ and let $Y$ be an $\mathbb{R}^k$ valued random variable with mean 0, covariance $\Sigma/n$ and $\|Y\| \leq \frac{\beta}{\sqrt{n}}$ almost surely. For $t \geq 0$, let $Z_t$ denote a Gaussian of mean 0 and covariance $t\Sigma$ independent of $Y$. Let $\sigma_{\min}^2$ denote the smallest eigenvalue of $\Sigma$. Then, for any $n \geq \frac{5\beta^2}{\sigma_{\min}^2}$, we have

$$\mathcal{W}_2(Z_1, Z_{1-1/n} + Y) \leq \frac{5\sqrt{k}\beta}{n\sqrt{n}}.$$

We note that the following Lemma is similar in form and proof to Lemma 7 of Kandasamy & Nagaraj (2024).

**Lemma 4.** Let $V$ be a random vector in $\mathbb{R}^d$ satisfying the following conditions:

1. $\|V\| \leq \beta$ a.s., $\mathbb{E}[V] = 0$, and $\mathbb{E}[VV^T] = \Sigma$.

2. $V$ lies in a one-dimensional subspace almost surely.

Suppose the random vector $Z$ is distributed as $\mathcal{N}(0, \mathbf{I}_d)$, and independent of $V$. Let $\nu = \mathrm{Tr}(\Sigma)$, Then,

$$\mathcal{W}_2^2(\mathrm{Law}(\sqrt{\mathbf{I}_d + \Sigma}Z), \mathrm{Law}(Z + V)) \leq 5\nu^2 + \mathbf{1}_{5\beta^2 > 1} \cdot 2\nu.$$

*Proof.* These distributions are the same along all directions perpendicular to $V$. We couple those directions identically. Let $V'$ denote the projection of $V$ onto the direction spanned by itself, and $Z'$ denote a one-dimensional Gaussian. We get

$$\mathcal{W}_2(\mathrm{Law}(\sqrt{\mathbf{I}_d + \Sigma}Z), \mathrm{Law}(Z + V)) \leq \mathcal{W}_2(\mathrm{Law}(\sqrt{1 + \nu}Z'), \mathrm{Law}(Z' + V'))$$
$$= \sqrt{1 + \nu}\mathcal{W}_2(\mathrm{Law}(Z'), \mathrm{Law}(\tfrac{Z'}{\sqrt{1+\nu}} + \tfrac{V'}{\sqrt{1+\nu}})).$$

Now set $k = 1$, $n = 1 + \frac{1}{\nu}$, and $\beta \to \beta\sqrt{n}$. Here $\sigma_{\min} = 1$, which means $5\beta^2 \leq 1$ is sufficient to apply Lemma 3.

$$\mathbf{1}_{5\beta^2 \leq 1} \cdot \mathcal{W}_2^2(\mathrm{Law}(\sqrt{\mathbf{I}_d + \Sigma}Z), \mathrm{Law}(Z + V)) \leq \mathbf{1}_{5\beta^2 \leq 1} \cdot \frac{25\beta^2\nu^2}{1 + \nu} \leq 5\nu^2.$$

When $5\beta^2 > 1$, we couple $\mathrm{Law}(\sqrt{1 + \nu}Z')$ and $\mathrm{Law}(Z' + V')$ to have the same Gaussian noise $Z'$, with $V'$ sampled independently of $Z'$. A simple computation yields

$$\mathbf{1}_{5\beta^2 > 1} \cdot \mathcal{W}_2^2(\mathrm{Law}(\sqrt{1 + \nu}Z'), \mathrm{Law}(Z' + V')) \leq \mathbf{1}_{5\beta^2 > 1} \cdot 2\nu.$$

$\square$

## C   PROOF FOR OVERDAMPED PLMC

Recall from Section 2.3 that $X_{t,i}$ denote the iterates of overdamped Langevin Monte Carlo with step-size $\frac{\eta}{k}$. Similarly $\tilde{X}_{t,i}$ denote the iterates of Poisson overdamped Langevin Monte Carlo with step size $\frac{\eta}{k}$, and $\tilde{X}_{t,i}^+$ denote midpoints.

$$X_{t,i+1} = X_{t,i} - \frac{\eta}{k}\nabla f(X_{t,i}) + \sqrt{\frac{2\eta}{k}}Y_{t,i}$$

$$\tilde{X}_{t,i+1} = \tilde{X}_{t,i} - \frac{\eta}{k}\nabla f(\tilde{X}_{t,0}) + \eta H_{t,i}(\nabla f(\tilde{X}_{t,0}) - \nabla f(\tilde{X}_{t,i}^+)) + \sqrt{\frac{2\eta}{k}}Z_{t,i}$$

$$\tilde{X}_{t,i}^+ = \tilde{X}_{t,0} - \frac{\eta i}{k}\nabla f(\tilde{X}_{t,0}) + \sqrt{\frac{2\eta}{k}}\sum_{j=0}^{i}Z_{t,j}$$

The sequences $Z_{t,i}$ and $Y_{t,i}$ are i.i.d. standard Gaussians, and $H_{t,i}$ are independent Bernoullis with parameter $1/k$. All random variables above live on the same probability space, with a coupling we will specify. To interpret PLMC as LMC with a perturbed noise, we write

$$\tilde{X}_{t,i+1} = \tilde{X}_{t,i} - \frac{\eta}{k}\nabla f(\tilde{X}_{t,i}) + \sqrt{\frac{2\eta}{k}}\tilde{Z}_{t,i},$$

where $\tilde{Z}_{t,i}$ denotes the perturbed Gaussian and is given by the following expression.

$$\tilde{Z}_{t,i} = \sqrt{\frac{\eta k}{2}}(H_{t,i} - 1/k)(\nabla f(\tilde{X}_{t,0}) - \nabla f(\tilde{X}_{t,i}^+)) + \sqrt{\frac{\eta}{2k}}(\nabla f(\tilde{X}_{t,i}) - \nabla f(\tilde{X}_{t,i}^+)) + Z_{t,i}.$$

Let $B_{t,i} = \sqrt{\frac{\eta}{2k}}(\nabla f(\tilde{X}_{t,i}) - \nabla f(\tilde{X}_{t,i}^+))$, and $S_{t,i} = \sqrt{\frac{\eta k}{2}}(H_{t,i} - 1/k)(\nabla f(\tilde{X}_{t,0}) - \nabla f(\tilde{X}_{t,i}^+))$. We refer to these as the bias and variance terms respectively.

Define the event:
$$\mathcal{G} = \{\tilde{X}_{t,0} = y_0, \tilde{X}_{t,i} = y, \tilde{X}_{t,i}^+ = y^+, X_{t,i} = x\},$$

with $x, y, y^+$ and $y_0$ being arbitrary points in $\mathbb{R}^d$. For any valid coupling of $X_{t,i+1}$ and $\tilde{X}_{t,i+1}$ conditioned on $\mathcal{G}$, the following holds.

**Proposition 1.** Let Assumption 1 hold and let $\frac{\alpha\eta}{k} < 1$. Then we have,

$$\mathbb{E}[||X_{t,i+1} - \tilde{X}_{t,i+1}||^2|\mathcal{G}] \leq (1 - \frac{\alpha\eta}{2k})^2||x-y||^2 + \frac{9\eta L^2}{\alpha k}||y-y^+||^2 + \frac{2\eta}{k}\mathbb{E}[||Z_{t,i} + S_{t,i} - Y_{t,i}||^2|\mathcal{G}].$$

The proof of this Proposition is in Section D.1. The first term arises from the contractivity of gradient descent, while the second term comes from the bias. We apply Lemma 1 to bound the final term.

**Corollary 3.** Let $\nu = \text{Tr}(S_{t,i}S_{t,i}^T|\mathcal{G})$, and $\beta^2 = \frac{\eta k L^2}{2}||y_0 - y^+||^2$. Let $\mathcal{E} \in \sigma(\tilde{X}_{t,0}, \tilde{X}_{t,i}, \tilde{X}_{t,i}^+, X_{t,i})$ be an event. Conditioned on $\mathcal{G}$, there exists a coupling of $Y_{t,i}, H_{t,i}$ and $Z_{t,i}$ such that under Assumption 1,

$$\mathbb{E}[||Z_{t,i} + S_{t,i} - Y_{t,i}||^2|\mathcal{G}] \leq (\mathbf{1}_{\mathcal{E}} + \mathbf{1}_{5\beta^2 > 1}) \cdot 2\nu + \mathbf{1}_{\mathcal{E}^c} \cdot \frac{11}{2}\nu^2.$$

*Proof.* Under the event $\mathcal{E}$, we couple the Gaussians $Y_{t,i}$ and $Z_{t,i}$ identically (i.e, $Y_{t,i} = Z_{t,i}$). This gives $\mathbb{E}[||Z_{t,i} + S_{t,i} - Y_{t,i}||^2|\mathcal{G}] = \mathbb{E}[||S_{t,i}||^2|\mathcal{G}] = \nu$. Under $\mathcal{E}^c$, couple them as in the Lemma 1. $\square$

**Remark 6.** Note that $\mathbb{E}(H_{t,i} - 1/k)^2 \leq 1/k$, so $\nu \leq \eta^2 L^2||\tilde{X}_{t,0} - \tilde{X}_{t,i}^+||^2$. The above Corollary is a slight technical modification of Lemma 1. We later choose $\mathcal{E}$ so that we may neglect terms proportional to $||\nabla f(\tilde{X}_{t,0})||^4$, arising from our bounds on $\nu^2$. This is detailed in Lemma 7.

With the above results, we produce an explicit coupling of $X_{t,i+1}$ and $\tilde{X}_{t,i+1}$ to bound the Wasserstein distance between their distributions. This involves coupling $X_{t,i}$ optimally with $\tilde{X}_{t,i}$, and bounding movement terms of the form $\mathbb{E}||\tilde{X}_{t,i} - \tilde{X}_{t,0}||^p$ and $\mathbb{E}||\tilde{X}_{t,i}^+ - \tilde{X}_{t,0}||^p$. These moments can be reduced to gradient and Gaussian terms, using the following Lemma.

**Lemma 5** (Lemma 12, Kandasamy & Nagaraj (2024)). Let $M_{t,k} = \sup_{0 \leq j < k}||\sum_{i=0}^{j}\sqrt{\frac{2\eta}{k}}Z_{t,i}||$, and $p \in \mathbb{N}$. Let $N_t := \sum_{i=0}^{k-1}H_{t,i}$. Then the following bounds are true.

$$\sup_{0 \leq i \leq k-1}||\tilde{X}_{t,i}^+ - \tilde{X}_{t,0}|| \leq \eta||\nabla f(\tilde{X}_{t,0})|| + M_{t,k}.$$

$$\sup_{0 \leq i \leq k-1}||\tilde{X}_{t,i}^+ - \tilde{X}_{t,i}|| \leq \eta L N_t \sup_{i \leq k-1}||\tilde{X}_{t,i}^+ - \tilde{X}_{t,0}||.$$

$$\mathbb{E}[M_{t,k}^p] \leq (\eta d)^{p/2}.$$

The following Lemma is proven in Section D.2.

**Lemma 6.** Assume $\eta L/k \leq 1$, and Assumption 1. Then there exist absolute constants $c_1, c_2 > 0$ such that

$$\mathcal{W}_2^2(\text{Law}(X_{t,i+1}), \text{Law}(\tilde{X}_{t,i+1})) \leq (1 - \frac{\alpha\eta}{2k})\mathcal{W}_2^2(\text{Law}(\tilde{X}_{t,i}), \text{Law}(X_{t,i})) + E_{t,i}, \text{ where}$$

$$E_{t,i} \lesssim \left(\eta^6 L^4 d + \frac{\eta^4 L^2}{k} + \frac{\eta^5 L^4}{\alpha k}\right)\mathbb{E}||\nabla f(\tilde{X}_{t,0})||^2$$
$$+ \frac{\eta^4 L^4 d}{\alpha k} + \frac{\eta^5 L^4 d^2}{k} + \exp(c_1 d - (c_2\eta^2 L^2 k)^{-1}) \cdot \frac{\eta^3 L^2 d}{k}.$$

**Finishing the proof.** Open the recursion in Lemma 6, summing the constant terms as a geometric series.

$$\mathcal{W}_2^2(X_{t,0}, \tilde{X}_{t,0}) \lesssim \exp(-\alpha\eta t)\mathcal{W}_2^2(X_{0,0}, \tilde{X}_{0,0})^2 + (\eta^6 L^4 kd + \eta^4 L^2 + \frac{\eta^5 L^4}{\alpha})\sum_{s=0}^{t-1}\mathbb{E}||\nabla f(\tilde{X}_{s,0})||^2$$

$$+ \frac{\eta^3 L^4 d}{\alpha^2} + \frac{\eta^4 L^4 d^2}{\alpha} + \exp(c_1 d - (c_2\eta^2 L^2 k)^{-1}) \cdot \frac{\eta^2 L^2 d}{\alpha}.$$

Note that $X_{0,0} = \tilde{X}_{0,0}$, so $\mathcal{W}_2^2(X_{0,0}, \tilde{X}_{0,0}) = 0$. The gradient term $\sum_{t=0}^{N-1}\mathbb{E}||\nabla f(\tilde{X}_{t,0})||^2$ is bounded in the following Lemma 2, proven in Section D.3.

# D    DEFERRED PROOFS FOR OVERDAMPED PLMC

## D.1    PROOF OF PROPOSITION 1

Let $T(x) = x - \frac{\eta}{k}\nabla f(x)$. Under the assumption $\alpha\eta/k < 1$, it follows from the strong convexity and smoothness of $f$ that $T$ is a contraction with Lipschitz constant $(1 - \frac{\alpha\eta}{k})$. By definition, we have

$$X_{t,i+1} = T(X_{t,i}) + \sqrt{\frac{2\eta}{k}}Y_{t,i}, \text{ and } \tilde{X}_{t,i+1} = T(\tilde{X}_{t,i}) + \sqrt{\frac{2\eta}{k}}\tilde{Z}_{t,i}.$$

Under the event $\mathcal{G}$, we have:

$$\|X_{t,i+1} - \tilde{X}_{t,i+1}\|^2 = \|T(x) - T(y)\|^2 + \frac{2\eta}{k}\|Y_{t,i} - \tilde{Z}_{t,i}\|^2$$
$$+ 2\sqrt{\frac{2\eta}{k}}\langle Y_{t,i} - \tilde{Z}_{t,i}, T(x) - T(y)\rangle$$

Conditioned on $\mathcal{G}$, $(H_{t,i} - 1/k)$ has zero mean, and $Y_{t,i}, Z_{t,i}$ are standard Gaussians. This leads to

$$\mathbb{E}[\|X_{t,i+1} - \tilde{X}_{t,i+1}\|^2|\mathcal{G}] = \|T(x) - T(y)\|^2 - \frac{2\eta}{k}\langle\nabla f(y) - \nabla f(y^+), T(x) - T(y)\rangle$$
$$+ \frac{2\eta}{k}\mathbb{E}[\|Y_{t,i} - \tilde{Z}_{t,i}\|^2|\mathcal{G}]$$
$$\leq (1 - \frac{\alpha\eta}{k})^2\|x - y\|^2 + \frac{2\eta L}{k}(1 - \frac{\alpha\eta}{k})\|y - y^+\| \cdot \|x - y\|$$
$$+ \frac{2\eta}{k}\mathbb{E}[\|Y_{t,i} - \tilde{Z}_{t,i}\|^2|\mathcal{G}].$$

The second term is bounded using the AM-GM inequality. For any arbitrary $\gamma > 0$,

$$\frac{2\eta L}{k}\|y - y^+\| \cdot \|x - y\| \leq \frac{4\eta^2 L^2}{\gamma}\|y - y^+\|^2 + \frac{\gamma}{k^2}\|x - y\|^2.$$

In particular, with $\gamma = \alpha\eta k/2$,

$$(1 - \frac{\alpha\eta}{k})^2\|x - y\|^2 + \frac{2\eta L}{k}(1 - \frac{\alpha\eta}{k})\|y - y^+\| \cdot \|x - y\|$$
$$\leq (1 - \frac{\alpha\eta}{k})(1 - \frac{\alpha\eta}{2k})\|x - y\|^2 + (1 - \frac{\alpha\eta}{k})\frac{8\eta L^2}{\alpha k}\|y - y^+\|^2$$
$$\leq (1 - \frac{\alpha\eta}{2k})^2\|x - y\|^2 + \frac{8\eta L^2}{\alpha k}\|y - y^+\|^2.$$

By definition of $\tilde{Z}_{t,i}$,

$$\tilde{Z}_{t,i} - Y_{t,i} = \sqrt{\frac{\eta}{2k}}(\nabla f(y) - \nabla f(y^+)) + Z_{t,i} + S_{t,i} - Y_{t,i}.$$

Square both sides, noting that $\mathbb{E}[Z_{t,i} + S_{t,i} - Y_{t,i}|\mathcal{G}] = 0$, and $\|\nabla f(y) - \nabla f(y^+)\|^2 \leq L^2\|y - y^+\|^2$ under assumption 1. This gives

$$\frac{2\eta}{k}\mathbb{E}[\|Y_{t,i} - \tilde{Z}_{t,i}\|^2|\mathcal{G}] = \frac{\eta^2 L^2}{k^2}\|y - y^+\|^2 + \frac{2\eta}{k}\mathbb{E}[\|Z_{t,i} + S_{t,i} - Y_{t,i}\|^2|\mathcal{G}]$$
$$\leq \frac{\eta L^2}{\alpha k}\|y - y^+\|^2 + \frac{2\eta}{k}\mathbb{E}[\|Z_{t,i} + S_{t,i} - Y_{t,i}\|^2|\mathcal{G}].$$

## D.2    PROOF OF LEMMA 6

*Proof.* Recall the definition $\mathcal{G} := \{\tilde{X}_{t,0} = y_0, \tilde{X}_{t,i} = y, \tilde{X}_{t,i}^+ = y^+, X_{t,i} = x\}$. Conditioned on $\mathcal{G}$, we have:

$$X_{t,i+1} = x - \frac{\eta}{k}\nabla f(x) + \sqrt{\frac{2\eta}{k}}Y_{t,i}$$

$$\tilde{X}_{t,i+1} = y + \eta H_{t,i}(\nabla f(y_0) - \nabla f(y^+)) - \frac{\eta}{k}\nabla f(y_0) + \sqrt{\frac{2\eta}{k}}Z_{t,i}.$$

Conditioned on $\mathcal{G}$, we couple $(Z_{t,i}, H_{t,i})$ and $Y_{t,i}$ as in Corollary 3. This allows us to define $(X_{t,i+1}, \tilde{X}_{t,i+1})$ using the equations above and gives a conditional coupling of $(Y_{t,i}, H_{t,i}, Z_{t,i}, X_{t,i+1}, \tilde{X}_{t,i+1})$.

We produce an unconditional coupling as follows: Couple $X_{t,i}$ and $\tilde{X}_{t,i}$ optimally w.r.t. to $\mathcal{W}_2$, then sample $\tilde{X}_{t,i}^+$ and $\tilde{X}_{t,0}$ jointly conditioned on $\tilde{X}_{t,i}$. Conditioned on $(\tilde{X}_{t,i}^+, \tilde{X}_{t,0}, X_{t,i}, \tilde{X}_{t,i})$ (i.e, $\sigma(\tilde{X}_{t,0}, \tilde{X}_{t,i}, \tilde{X}_{t,i}^+, X_{t,i})$), we then sample $(Z_{t,i}, Y_{t,i}, H_{t,i}, X_{t,i+1}, \tilde{X}_{t,i+1})$ from the conditional coupling described above. Taking the expectation in Proposition 1, and using the bounds in Corollary 3 we get:

$$\mathcal{W}_2^2(X_{t,i+1}, \tilde{X}_{t,i+1})^2 \le (1 - \frac{\alpha\eta}{2k})^2\mathcal{W}_2^2(X_{t,i}, \tilde{X}_{t,i})^2 + E_{t,i},$$

where $E_{t,i} \lesssim \frac{\eta L^2}{\alpha k}\mathbb{E}||\tilde{X}_{t,i} - \tilde{X}_{t,i}^+||^2 + \frac{\eta}{k}\mathbb{E}[(\mathbf{1}_{\mathcal{E}} + \mathbf{1}_{5\beta^2>1}) \cdot 2\nu + \mathbf{1}_{\mathcal{E}^c} \cdot \frac{11}{2}\nu^2]$ and $\mathcal{E} \in \sigma(\tilde{X}_{t,0}, \tilde{X}_{t,i}, \tilde{X}_{t,i}^+, X_{t,i})$ is any event. We choose a particular event $\mathcal{E}$ and bound the latter term in Lemma 7. The former term is bounded below, using items 1 and 2 of Lemma 5.

$$\frac{\eta L^2}{\alpha k}\mathbb{E}||\tilde{X}_{t,i} - \tilde{X}_{t,i}^+||^2 \lesssim \frac{\eta^3 L^4}{\alpha k}\mathbb{E}\left[N_t^2 \sup_{j \le k-1}||\tilde{X}_{t,0} - \tilde{X}_{t,j}^+||^2\right]$$

Note that $N_t$ is independent of $y_0$ and $y^+$, and $\mathbb{E}[N_t^2] \lesssim 1$. Along with item 2 of Lemma 5, this gives

$$\frac{\eta L^2}{\alpha k}\mathbb{E}||\tilde{X}_{t,i} - \tilde{X}_{t,i}^+||^2 \lesssim \frac{\eta^5 L^4}{\alpha k}\mathbb{E}||\nabla f(\tilde{X}_{t,0})||^2 + \frac{\eta^3 L^4}{\alpha k}\mathbb{E}[M_{t,k}^2]$$
$$\lesssim \frac{\eta^5 L^4}{\alpha k}\mathbb{E}||\nabla f(\tilde{X}_{t,0})||^2 + \frac{\eta^4 L^4 d}{\alpha k}.$$

$\square$

### D.3    Proof of Lemma 2

*Proof.* Since $f$ is smooth, we have (Lemma 3.4, Bubeck et al. (2015))

$$f(\tilde{X}_{t+1,0}) - f(\tilde{X}_{t,0}) \le \langle \nabla f(\tilde{X}_{t,0}), \tilde{X}_{t+1,0} - \tilde{X}_{t,0}\rangle + \frac{L}{2}||\tilde{X}_{t+1,0} - \tilde{X}_{t,0}||^2.$$

By definition, $\tilde{X}_{t+1,0} - \tilde{X}_{t,0} = -\eta\nabla f(\tilde{X}_{t,0}) + \sum_{i=0}^{k-1}\eta H_{t,i}(\nabla f(\tilde{X}_{t,0}) - \nabla f(\tilde{X}_{t,i}^+)) + \sum_{i=0}^{k-1}\sqrt{\frac{2\eta}{k}}Z_{t,i}$. Since $\mathbb{E}[H_{t,i}] = 1/k$ and $\mathbb{E}[Z_{t,i}] = 0$,

$$\mathbb{E}\langle \nabla f(\tilde{X}_{t,0}), \tilde{X}_{t+1,0} - \tilde{X}_{t,0}\rangle \le -\eta\mathbb{E}||\nabla f(\tilde{X}_{t,0})||^2$$
$$+ \sum_{i=0}^{k-1}\frac{\eta}{k}\mathbb{E}||\nabla f(\tilde{X}_{t,0})|| \cdot ||\nabla f(\tilde{X}_{t,0}) - \nabla f(\tilde{X}_{t,i}^+)||$$
$$\le -\frac{\eta}{2}\mathbb{E}||\nabla f(\tilde{X}_{t,0})||^2 + \sum_{i=0}^{k-1}\frac{\eta}{2k}\mathbb{E}||\nabla f(\tilde{X}_{t,0}) - \nabla f(\tilde{X}_{t,i}^+)||^2$$
$$\le -\frac{\eta}{2}\mathbb{E}||\nabla f(\tilde{X}_{t,0})||^2 + \frac{\eta}{2}\sup_{0 \le i \le k-1}\mathbb{E}||\nabla f(\tilde{X}_{t,0}) - \nabla f(\tilde{X}_{t,i}^+)||^2$$
$$\le -\frac{\eta}{2}\mathbb{E}||\nabla f(\tilde{X}_{t,0})||^2 + \eta^3 L^2||\nabla f(\tilde{X}_{t,0})||^2 + \eta^2 L^2 d.$$

Where in the second and final steps we used $ab \le \frac{a^2+b^2}{2}$ and Lemma 5 respectively. Now we use $||a+b||^2 \le 2(||a||^2 + ||b||^2)$ and $\mathbb{E}||\sum_{i=0}^{k-1}\sqrt{\frac{2\eta}{k}}Z_{t,i}||^2 = 2\eta d$ to get

$$\frac{L}{2}||\tilde{X}_{t+1,0} - \tilde{X}_{t,0}||^2 \le \eta^2 L||\nabla f(\tilde{X}_{t,0})||^2 + \eta^2 L||\sum_{i=0}^{k-1}H_{t,i}(\nabla f(\tilde{X}_{t,0}) - \nabla f(\tilde{X}_{t,i}^+))||^2 + 2\eta L d.$$

Let $N_t = \sum_{i=0}^{k-1} H_{t,i}$. Note that $\mathbb{E}[N_t^2] \leq 2$, and $N_t$ is independent of $\tilde{X}_{t,0}$. Triangle inequality and 5 give

$$\eta^2 L \mathbb{E}\| \sum_{i=0}^{k-1} H_{t,i}(\nabla f(\tilde{X}_{t,0}) - \nabla f(\tilde{X}_{t,i}^+))\|^2 \leq \eta^2 L \mathbb{E}[N_t \sup_{0 \leq i \leq k-1} \mathbb{E}\|\nabla f(\tilde{X}_{t,0}) - \nabla f(\tilde{X}_{t,i}^+)\|]^2$$
$$\leq 4\eta^4 L^3 \mathbb{E}\|\nabla f(\tilde{X}_{t,0})\|^2 + 4\eta^3 L^3 d.$$

Under our assumption $\eta L \leq 1/8$, the terms $\eta^3 L^2 \|\nabla f(\tilde{X}_{t,0})\|^2, \eta^4 L^3 \mathbb{E}\|\nabla f(\tilde{X}_{t,0})\|^2, \eta^2 L^2 d$ and $\eta^3 L^3 d$ are negligible in order. Collecting the dominant terms, we get

$$\eta \mathbb{E}\|\nabla f(\tilde{X}_{t,0})\|^2 \lesssim [f(\tilde{X}_{t,0}) - f(\tilde{X}_{t+1,0})] + \eta L d.$$

This telescopes, leading to the result. $\qquad \square$

## D.4 PROOF OF COROLLARY 1

*Proof.* By triangle inequality on $\mathcal{W}_2$,

$$\mathcal{W}_2^2(\text{Law}(\tilde{X}_{N,0}), \pi) \lesssim \mathcal{W}_2^2(\text{Law}(\tilde{X}_{N,0}), \text{Law}(X_{N,0})) + \mathcal{W}_2^2(\text{Law}(X_{N,0}), \pi).$$

We show under the conditions of our Corollary that each of these terms is $\mathcal{O}(\epsilon^2 d/\alpha)$. To deal with the second term, recall the following Theorem for the convergence of Langevin Monte-Carlo.

**Theorem 3** (Corollary 10, Durmus et al. (2019)). Suppose Assumption 1 is true. Let $X_n$ denote the iterates of Langevin Monte-Carlo with step-size $\gamma_\epsilon$. Then, with

$$\gamma_\epsilon = \frac{\epsilon^2}{4L}, \quad n_\epsilon \geq \lceil \log(\frac{2\mathcal{W}_2^2(X_0, \pi)\alpha}{\epsilon^2 d})\gamma_\epsilon^{-1}\alpha^{-1} \rceil$$

we have $\mathcal{W}_2^2(X_{n_\epsilon}, \pi) \leq \frac{\epsilon^2 d}{\alpha}$.

By our choice of $k$, we have $\frac{\eta}{k} \lesssim \frac{\epsilon^2}{L}$. Note that the above Theorem goes through with an inequality $\eta \leq \frac{\epsilon^2}{4L}$, so we have $\mathcal{W}_2^2(X_{N,0}, \pi) \leq \frac{\epsilon^2 d}{\alpha}$ for $N = \log(\frac{2\mathcal{W}_2^2(X_{0,0}, \pi)\alpha}{\epsilon^2 d})(\alpha\eta)^{-1}$. Let $L_1 = 2\max(C_f, \log(\frac{2\mathcal{W}_2^2(X_{0,0}, \pi)\alpha}{\epsilon^2 d}))$. Now apply Theorem 1 with

$$\eta \asymp \min\Big( \frac{\epsilon^{2/3}}{L_1^{1/3}L}, \frac{\epsilon^{1/2}}{\kappa^{1/4}L_1^{1/4}L}, \frac{\epsilon^{2/3}}{d^{1/6}L_1^{1/6}L}, \frac{\epsilon^{2/3}}{\kappa^{1/3}L}, \frac{\epsilon^{1/2}}{d^{1/4}L}, \big(\frac{c_2\epsilon^2}{c_1 d - \log \epsilon^2}\big)^{1/3} \cdot \frac{1}{L} \Big)$$

and $N$ as above, to see $\mathcal{W}_2^2(\text{Law}(\tilde{X}_{N,0}), \text{Law}(X_{N,0})) \lesssim \frac{\epsilon^2 d}{\alpha}$. $\qquad \square$

## E TECHNICAL RESULTS FOR OLMC

**Lemma 7.** Let $\beta$ and $\nu$ be defined as in Lemma 3. Define the event $\mathcal{E} \in \sigma(\tilde{X}_{t,0}, \tilde{X}_{t,i}, \tilde{X}_{t,i}^+, X_{t,i})$ by $\mathcal{E} = \{\frac{\eta^4 L^2}{k}\|\nabla f(\tilde{X}_{t,0})\|^2 < \frac{\eta^7 L^4}{k}\|\nabla f(\tilde{X}_{t,0})\|^4\}$. Then

$$\frac{\eta}{k}\mathbb{E}[(\mathbf{1}_{\mathcal{E}} + \mathbf{1}_{5\beta^2 > 1}) \cdot \nu + \mathbf{1}_{\mathcal{E}^c} \cdot \nu^2] \lesssim (\eta^6 L^4 d + \frac{\eta^4 L^2}{k})\mathbb{E}\|\nabla f(\tilde{X}_{t,0})\|^2 + \frac{\eta^5 L^4 d^2}{k}$$
$$+ \exp(c_1 d - (c_2 \eta^2 L^2 k)^{-1}) \cdot \frac{\eta^3 L^2 d}{k}$$

where the expectation is taken over the distribution defined in the proof of 6.

*Proof.* Since $H_{t,i}$ is a Bernoulli random variable with parameter $1/k$, we have $\mathbb{E}[(H_{t,i} - 1/k)^2] \leq 1/k$. This gives us an upper bound on $\nu$, since $\nu = \mathbb{E}[\frac{\eta k}{2}(H_{t,i} - 1/k)^2 \|\nabla f(y_0) - \nabla f(y^+)\|^2] \leq \frac{\eta L^2}{2}\|y_0 - y^+\|^2$ under Assumption 1. This gives

$$\frac{\eta}{k}[(\mathbf{1}_{\mathcal{E}} + \mathbf{1}_{5\beta^2 > 1}) \cdot \nu + \mathbf{1}_{\mathcal{E}^c} \cdot \nu^2] \lesssim (\mathbf{1}_{\mathcal{E}} + \mathbf{1}_{5\beta^2 > 1}) \cdot \frac{\eta^2 L^2}{k}\|y_0 - y^+\|^2 + \mathbf{1}_{\mathcal{E}^c} \cdot \frac{\eta^3 L^4}{k}\|y_0 - y^+\|^4$$

Now we apply item 1 of Lemma 5 to obtain the following.

$$\frac{\eta^2 L^2}{k}\|y_0 - y^+\|^2 \lesssim \frac{\eta^4 L^2}{k}\|\nabla f(\tilde{X}_{t,0})\|^2 + \frac{\eta^2 L^2}{k}M_{t,k}^2.$$

$$\frac{\eta^3 L^4}{k}\|y_0 - y^+\|^4 \lesssim \frac{\eta^7 L^4}{k}\|\nabla f(\tilde{X}_{t,0})\|^4 + \frac{\eta^3 L^4}{k}M_{t,k}^4.$$

Using $(\mathbf{1}_{5\beta^2 > 1} + \mathbf{1}_{\mathcal{E}}) \lesssim 1$ and $\mathbf{1}_{\mathcal{E}^c} \frac{\eta^7 L^4}{k}\|\nabla f(\tilde{X}_{t,0})\|^4 \leq \frac{\eta^4 L^2}{k}\|\nabla f(\tilde{X}_{t,0})\|^2$, we obtain

$$\frac{\eta}{k}[(\mathbf{1}_{\mathcal{E}} + \mathbf{1}_{5\beta^2 > 1}) \cdot \nu + \mathbf{1}_{\mathcal{E}^c} \cdot \nu^2] \lesssim \frac{\eta^4 L^2}{k}\|\nabla f(\tilde{X}_{t,0})\|^2 + \frac{\eta^3 L^4}{k}M_{t,k}^4$$
$$+ (\mathbf{1}_{\mathcal{E}} + \mathbf{1}_{5\beta^2 > 1})\frac{\eta^2 L^2}{k}M_{t,k}^2.$$

The expectations of the second term and final terms are bounded in Lemmas 5 and 8 respectively. □

**Lemma 8.** Let $\beta$ and $\mathcal{E}$ be as in Lemma 7. There exists an absolute constants $c_1$ and $c_2$ such that

$$\mathbb{E}[(\mathbf{1}_{5\beta^2 > 1} + \mathbf{1}_{\mathcal{E}}) \cdot \frac{\eta^2 L^2}{k}M_{t,k}^2] \lesssim \eta^6 L^4 d\mathbb{E}\|\nabla f(\tilde{X}_{t,0})\|^2 + \exp(c_1 d - (c_2 \eta^2 L^2 k)^{-1})\frac{\eta^3 L^2 d}{k}$$

*Proof.* Note that $\mathcal{E}$ is independent of $M_{t,k}$, and by its definition we have $1_{\mathcal{E}} \leq \eta^3 L^2\|\nabla f(\tilde{X}_{t,0})\|^2$. As a result,

$$\mathbb{E}[1_{\mathcal{E}} \cdot \frac{\eta^2 L^2}{k}M_{t,k}^2] \leq \eta^3 L^2 \mathbb{E}\|\nabla f(\tilde{X}_{t,0})\|^2 \cdot \mathbb{E}[\frac{\eta^2 L^2}{k}M_{t,k}^2].$$

Recall the definition of $\beta$.

$$\beta \leq \sqrt{\eta k}L\|\tilde{X}_{t,0} - \tilde{X}_{t,i}^+\|$$
$$= \sqrt{\eta k}L\left\|\frac{\eta i}{k}\nabla f(\tilde{X}_{t,0}) + \sqrt{\frac{2\eta}{k}}\sum_{j=0}^{i} Z_{t,j}\right\|.$$

Applying triangle inequality and union bound, we get

$$\mathbf{1}_{\sqrt{5}\beta > 1} \leq \mathbf{1}\{\sqrt{5}\eta^{3/2}k^{1/2}L\|\nabla f(\tilde{X}_{t,0})\| > 1\} + \mathbf{1}\{\sqrt{10}\eta L\|\sum_{j=0}^{i} Z_{t,j}\| > 1\}.$$

Note that $\tilde{X}_{t,0}$ is independent of $M_{t,k}$. To handle the second term below, apply Cauchy Schwarz and a Gaussian concentration inequality.

$$\mathbb{E}[\mathbf{1}_{5\beta^2 > 1} \cdot \frac{\eta^2 L^2}{k}M_{t,k}^2] \leq \mathbb{P}[\sqrt{5}\eta^{3/2}k^{1/2}L\|\nabla f(\tilde{X}_{t,0})\| > 1] \cdot \frac{\eta^2 L^2}{k}\mathbb{E}[M_{t,k}^2]$$
$$+ \mathbb{P}[\sqrt{10}\eta L\|\sum_{j=0}^{i} Z_{t,j}\| > 1]^{1/2} \cdot \frac{\eta^2 L^2}{k}\mathbb{E}[M_{t,k}^4]^{1/2}$$
$$\lesssim \eta^3 k L^2 \mathbb{E}[\|\nabla f(\tilde{X}_{t,0})\|^2] \cdot \frac{\eta^2 L^2}{k}\mathbb{E}[M_{t,k}^2]$$
$$+ \exp(c_1 d - (c_2 \eta^2 L^2 k)^{-1}) \cdot \frac{\eta^2 L^2}{k}\mathbb{E}[M_{t,k}^4]^{1/2}.$$

Where $c_1, c_2 > 0$ are absolute constants. Applying item 2 of Lemma 5 completes the proof. □

## F    PROOF FOR UNDERDAMPED PLMC

### F.1    BASIS CHANGE FOR CONTRACTIVITY

Recall from Section 2.3 the definitions of $\tilde{U}_{t,i}, \tilde{V}_{t,i}$. We make the following coordinate change for the iterates of underdamped LMC/PLMC.

$$\begin{bmatrix} x \\ y \end{bmatrix} \to \mathcal{M} \begin{bmatrix} x \\ y \end{bmatrix}, \text{ where } \mathcal{M} = \begin{bmatrix} \mathbf{I}_d & 0 \\ \mathbf{I}_d & \frac{2}{\gamma}\mathbf{I}_d \end{bmatrix}.$$

We denote $W_{t,i} = U_{t,i} + \frac{2}{\gamma}V_{t,i}$, and $\tilde{W}_{t,i} = \tilde{U}_{t,i} + \frac{2}{\gamma}\tilde{V}_{t,i}$. Similarly, $\tilde{W}_{t,i}^+ = \tilde{U}_{t,i}^+ + \frac{2}{\gamma}\tilde{V}_{t,i}^+$, and

$$\tilde{X}_{t,i} = \begin{bmatrix} \tilde{U}_{t,i} \\ \tilde{W}_{t,i} \end{bmatrix}, \tilde{X}_{t,i}^+ = \begin{bmatrix} \tilde{U}_{t,i}^+ \\ \tilde{W}_{t,i}^+ \end{bmatrix}, \text{ and } X_{t,i} = \begin{bmatrix} U_{t,i} \\ W_{t,i} \end{bmatrix}.$$

The transformed iterates $\tilde{U}_{t,i}, \tilde{W}_{t,i}$ satisfy the following recursion.

$$\begin{bmatrix} \tilde{U}_{t,i+1} \\ \tilde{W}_{t,i+1} \end{bmatrix} = A_{\mathcal{M}}\left(\frac{\eta}{k}\right) \begin{bmatrix} \tilde{U}_{t,i} \\ \tilde{W}_{t,i} \end{bmatrix} - G_{\mathcal{M}}\left(\frac{\eta}{k}\right) \begin{bmatrix} \nabla f(\tilde{U}_{t,0}) \\ 0 \end{bmatrix} + \Gamma_{\mathcal{M}}\left(\frac{\eta}{k}\right) Z_{t,i}$$
$$+ kH_{t,i} \cdot G_{\mathcal{M}}\left(\frac{\eta}{k}\right) \begin{bmatrix} \nabla f(\tilde{U}_{t,0}) - \nabla f(\tilde{U}_{t,i}^+) \\ 0 \end{bmatrix}$$

The matrices $A_{\mathcal{M}}, G_{\mathcal{M}}$ and $\Gamma_{\mathcal{M}}$ account for the change of basis. It can be verified that $A_{\mathcal{M}} = \mathcal{M}A\mathcal{M}^{-1}$, and $G_{\mathcal{M}} = \mathcal{M}G$. Moreover, $\Gamma_{\mathcal{M}} = \mathcal{M}\Gamma$, and these are explicated below.

$$A_{\mathcal{M}}(h) = \begin{bmatrix} \frac{1}{2}(1 + \exp(-\gamma h))\mathbf{I}_d & \frac{1}{2}(1 - \exp(-\gamma h))\mathbf{I}_d \\ \frac{1}{2}(1 - \exp(-\gamma h))\mathbf{I}_d & \frac{1}{2}(1 + \exp(-\gamma h))\mathbf{I}_d \end{bmatrix}, G_{\mathcal{M}}(h) = \begin{bmatrix} \frac{\gamma h - (1 - \exp(-\gamma h))}{\gamma^2}\mathbf{I}_d & 0 \\ \frac{\gamma h + (1 - \exp(-\gamma h))}{\gamma^2}\mathbf{I}_d & 0 \end{bmatrix}.$$

$$\Gamma_{\mathcal{M}}^2(h) = \begin{bmatrix} \frac{4(1 - \exp(-\gamma h)) - (1 - \exp(2\gamma h)) + 2\gamma h}{\gamma^2}\mathbf{I}_d & \frac{2\gamma h - (1 - \exp(2\gamma h))}{\gamma^2}\mathbf{I}_d \\ \frac{2\gamma h - (1 - \exp(2\gamma h))}{\gamma^2}\mathbf{I}_d & \frac{4(1 - \exp(-\gamma h)) + (1 - \exp(2\gamma h)) + 2\gamma h}{\gamma^2}\mathbf{I}_d \end{bmatrix}$$

In order to interpret this as ULMC with perturbed Gaussian noise, we write

$$\begin{bmatrix} \tilde{U}_{t,i+1} \\ \tilde{W}_{t,i+1} \end{bmatrix} = A_{\mathcal{M}}\left(\frac{\eta}{k}\right) \begin{bmatrix} \tilde{U}_{t,i} \\ \tilde{W}_{t,i} \end{bmatrix} + G_{\mathcal{M}}\left(\frac{\eta}{k}\right) \begin{bmatrix} -\nabla f(\tilde{U}_{t,i}) \\ 0 \end{bmatrix} + \Gamma_{\mathcal{M}}\left(\frac{\eta}{k}\right) \tilde{Z}_{t,i}.$$

The perturbed Gaussian $\tilde{Z}_{t,i}$ can be expressed as $\tilde{Z}_{t,i} = Z_{t,i} + B_{t,i} + S_{t,i}$, where

$$B_{t,i} = \Gamma_{\mathcal{M}}^{-1}\left(\frac{\eta}{k}\right) G_{\mathcal{M}}\left(\frac{\eta}{k}\right) \begin{bmatrix} \nabla f(\tilde{U}_{t,i}) - \nabla f(\tilde{U}_{t,i}^+) \\ 0 \end{bmatrix}$$

$$S_{t,i} = k(H_{t,i} - 1/k)\Gamma_{\mathcal{M}}\left(\frac{\eta}{k}\right)^{-1} G_{\mathcal{M}}\left(\frac{\eta}{k}\right) \begin{bmatrix} \nabla f(\tilde{U}_{t,0}) - \nabla f(\tilde{U}_{t,i}^+) \\ 0 \end{bmatrix}.$$

Here $B_{t,i}, S_{t,i}$ are called the bias and variance terms respectively.

The midpoints are given by

$$\begin{bmatrix} \tilde{U}_{t,i}^+ \\ \tilde{W}_{t,i}^+ \end{bmatrix} = A_{\mathcal{M}}\left(\frac{\eta i}{k}\right) \begin{bmatrix} \tilde{U}_{t,0} \\ \tilde{W}_{t,0} \end{bmatrix} - G_{\mathcal{M}}\left(\frac{\eta i}{k}\right) \begin{bmatrix} \nabla f(U_{t,0}) \\ 0 \end{bmatrix} + \sum_{j=0}^{i-1} A_{\mathcal{M}}\left(\frac{\eta(k-1-i)}{k}\right) G_{\mathcal{M}}\left(\frac{\eta i}{k}\right) Z_{t,i}.$$

The iterates of underdamped LMC satisfy

$$\begin{bmatrix} U_{t,i+1} \\ W_{t,i+1} \end{bmatrix} = A_{\mathcal{M}}\left(\frac{\eta}{k}\right) \begin{bmatrix} U_{t,i} \\ W_{t,i} \end{bmatrix} - G_{\mathcal{M}}\left(\frac{\eta}{k}\right) \begin{bmatrix} \nabla f(U_{t,i}) \\ 0 \end{bmatrix} + \Gamma_{\mathcal{M}}\left(\frac{\eta}{k}\right) Y_{t,i},$$

$$\begin{bmatrix} U_{t+1,0} \\ W_{t+1,0} \end{bmatrix} = \begin{bmatrix} U_{t,k} \\ W_{t,k} \end{bmatrix}.$$

Here $Y_{t,i}$ and $Z_{t,i}$ are i.i.d. standard Gaussians, $H_{t,i}$ are Bernoulli with parameter $1/k$, and all random variables above live on the same probability space with a coupling yet to be specified.

## F.2 PROOF OVERVIEW

Our proof follows the same method as in the overdamped case. As before, We condition on the previous iterates – with the following event:

$$\mathcal{G} = \left\{ \tilde{X}_{t,0} = y_0 = \begin{bmatrix} u_0 \\ w_0 \end{bmatrix}, \tilde{X}_{t,i} = y = \begin{bmatrix} \tilde{u} \\ \tilde{w} \end{bmatrix}, \tilde{X}_{t,i}^+ = y^+ = \begin{bmatrix} u^+ \\ w^+ \end{bmatrix}, X_{t,i} = x = \begin{bmatrix} u \\ w \end{bmatrix} \right\},$$

where $y_0, y, y^+$ and $x$ arbitrary points in $\mathbb{R}^{2d}$. For any valid coupling of $X_{t,i+1}$ and $\tilde{X}_{t,i+1}$, the following holds.

**Proposition 2.** Assume $\eta/k \lesssim \frac{1}{\kappa\sqrt{L}}$, and $\frac{\gamma\eta}{k} < c_0$ for sufficiently small $c_0 > 0$. Then with $\gamma = c_\gamma\sqrt{L}$ for some $c_\gamma \geq 2$, the following holds.

$$\mathbb{E}[\|X_{t,i+1} - \tilde{X}_{t,i+1}\|^2 | \mathcal{G}] \leq (1 - \Omega(\frac{\alpha\eta}{\gamma k}))\|x - y\|^2$$
$$+ O\left[\frac{\eta L^2}{\alpha\gamma k}\|u^+ - \tilde{u}\|^2 + \frac{\eta}{\gamma k}\mathbb{E}[\|Z_{t,i} + S_{t,i} - Y_{t,i}\|^2 | \mathcal{G}]\right].$$

The above Proposition is proved in Section I.1. The first term arises from the contractivity of the ULMC update rule, while the second term comes from the bias. Having conditioned on $\mathcal{G}$, we use 1 to bound the final term $\mathbb{E}[\|Z_{t,i} + S_{t,i} - Y_{t,i}\|^2 \mathcal{G}]$. We refer to Section G.1 for the proof of the following proposition.

**Proposition 3.** Let $p \geq 0$ be an integer. Conditioned on $\mathcal{G}$, there exists a coupling of $Z_{t,i}, H_{t,i}$ and $Y_{t,i}$ such that

$$\frac{\eta}{\gamma k}\mathbb{E}[\|Z_{t,i} + S_{t,i} - Y_{t,i}\|^2 | \mathcal{G}] \lesssim \frac{\eta^3 L^4}{\gamma^3 k}\|u_0 - u^+\|^4 + \frac{5^p \eta^{p+2} k^{p-1} L^{2p+2}}{\gamma^{p+2}}\|u_0 - u^+\|^{2p+2}.$$

**Remark 7.** The presence of $p$ is due to the manner in which handle the low probability event $\{5\beta^2 > 1\}$, appearing in Lemma 1. We use $\mathbf{1}_{5\beta^2 > 1} \leq 5^p\beta^{2p}$, with an appropriate bound on $\beta^{2p}$. Each choice of $p$ leads to a different error bound, so we write this in generality.

With the above results, we produce an explicit coupling of $X_{t,i+1}$ and $\tilde{X}_{t,i+1}$ to bound the Wasserstein distance between their distributions. This is done by coupling $X_{t,i}$ optimally with $\tilde{X}_{t,i}$, then bounding the moments $\mathbb{E}\|u^+ - \tilde{u}\|^2$ and $\mathbb{E}\|u_0 - \tilde{u}\|^p$. These moments contain gradient, momentum, and Gaussian terms; and are handled via the following Lemma.

**Lemma 9** (Lemma 21, Kandasamy & Nagaraj (2024)). Let $\Pi$ denote projection onto the position axis: $\Pi[u, v]^T = [u, 0]^T$. Let $M_{t,k} = \sup_{0 \leq i < k} \|\sum_{j=0}^{i} A(\frac{\eta(i-j)}{k})\Gamma(\frac{\eta}{k})Z_{t,j}\|_\Pi$. Then the following inequalities are true.

$$\|\tilde{U}_{t,i}^+ - \tilde{U}_{t,0}\| \lesssim \eta\|\tilde{V}_{t,0}\| + \eta^2\|\nabla f(\tilde{U}_{t,0})\| + M_{t,k}$$
$$\mathbb{E}[M_{t,k}^p] \lesssim \exp(\frac{p\gamma\eta}{2})\gamma^{p/2}\eta^{3p/2}(d + \log k)^{p/2}.$$

The proof of the following Lemma is given in Section G.2

**Lemma 10.** Assume $\eta/k \lesssim \frac{1}{\kappa\sqrt{L}}$, and $\gamma\eta < c_0$ for sufficiently small $c_0 > 0$. With $\gamma = c\sqrt{L}$ for some $c \geq 2$, the following is true.

$$\mathcal{W}_2^2(\text{Law}(X_{t,i+1}), \text{Law}(\tilde{X}_{t,i+1})) \leq (1 - \Omega(\frac{\alpha\eta}{\gamma k}))\mathcal{W}_2^2(X_{t,i}, \tilde{X}_{t,i}) + \mathcal{O}\left[\frac{\eta^7 L^4}{\alpha\gamma k}\mathbb{E}\|\tilde{V}_{t,0}\|^2\right.$$
$$+ \frac{\eta^9 L^4}{\alpha\gamma k}\mathbb{E}\|\nabla f(\tilde{U}_{t,0})\|^2 + \frac{\eta^8 L^4}{\alpha k}(d + \log k) + \frac{\eta^7 L^4}{\gamma^3 k}\mathbb{E}\|\tilde{V}_{t,0}\|^4$$
$$+ \frac{\eta^{11} L^4}{\gamma^3 k}\mathbb{E}\|\nabla f(\tilde{U}_{t,0})\|^4 + \frac{\eta^9 L^4}{\gamma k}(d + \log k)^2 + \lambda_p\left[\frac{\eta^{3p+4} k^{p-1} L^{2p+2}}{\gamma^{p+2}}\mathbb{E}\|\tilde{V}_{t,0}\|^{2p+2}\right.$$
$$\left.+ \frac{\eta^{5p+6} k^{p-1} L^{2p+2}}{\gamma^{p+2}}\mathbb{E}\|\nabla f(\tilde{U}_{t,0})\|^{2p+2} + \frac{\eta^{4p+5} k^{p-1} L^{2p+2}}{\gamma}(d + \log k)^{p+1}\right],$$

Where $\lambda_p$ is a constant depending only on $p$.

### F.3 FINISHING THE PROOF

Open up the recursion, summing up the constant terms as a geometric series. This gives

$$W_2^2(\text{Law}(\tilde{X}_{t,0}), \text{Law}(X_{t,0})) \lesssim \exp\left(\Omega(-\tfrac{\alpha\eta t}{\sqrt{L}})\right)\mathcal{W}_2^2(\text{Law}(X_{0,0}), \text{Law}(\tilde{X}_{0,0}))$$

$$+ \sum_{s=0}^{t} \left[\tfrac{\eta^7 L^4}{\alpha\gamma}\mathbb{E}\|\tilde{V}_{t,0}\|^2 + \tfrac{\eta^9 L^4}{\alpha\gamma}\mathbb{E}\|\nabla f(\tilde{U}_{t,0})\|^2\right] + \tfrac{\eta^7 L^4 \gamma}{\alpha^2}(d + \log k)$$

$$+ \sum_{s=0}^{t} \left[\tfrac{\eta^7 L^4}{\gamma^3}\mathbb{E}\|\tilde{V}_{t,0}\|^4 + \tfrac{\eta^{11} L^4}{\gamma^3}\mathbb{E}\|\nabla f(\tilde{U}_{t,0})\|^4\right] + \tfrac{\eta^8 L^4}{\alpha}(d + \log k)^2$$

$$+ \sum_{s=0}^{t} \lambda_p \left[\tfrac{\eta^{3p+4} k^p L^{2p+2}}{\gamma^{p+2}}\mathbb{E}\|\tilde{V}_{t,0}\|^{2p+2} + \tfrac{\eta^{5p+6} k^p L^{2p+2}}{\gamma^{p+2}}\mathbb{E}\|\nabla f(\tilde{U}_{t,0})\|^{2p+2}\right]$$

$$+ \lambda_p \tfrac{\eta^{4p+4} k^p L^{2p+2}}{\alpha}(d + \log k)^{p+1}$$

Note that $X_{0,0} = \tilde{X}_{0,0}$ by definition, so the first term is zero. The moments $\sum_{s=0}^{t} \mathbb{E}\|\tilde{V}_{t,0}\|^{2p}$ and $\sum_{s=0}^{t} \mathbb{E}\|\nabla f(\tilde{U}_{t,0})\|^{2p}$ are bounded the following Lemma.

**Theorem 4** (Theorem 4, Kandasamy & Nagaraj (2024)). Fix $p \geq 1$, and let $\mathcal{S}_{2p}(\nabla f) = \sum_{t=0}^{T} \mathbb{E}\|\nabla f(\tilde{U}_{t,0})\|^{2p}$. Let $\mathcal{S}_{2p}(V) = \sum_{t=0}^{T} \mathbb{E}\|\tilde{V}_{t,0}\|^{2p}$, and $\Psi_t = \tilde{U}_{t,0} + \frac{1}{\gamma}\tilde{V}_{t,0}$. There exist constants $C_p, c_p, \bar{c}_p > 0$ such that whenever: $\gamma \geq C_p\sqrt{L}$, $\alpha\gamma < c_p$, $\frac{\eta^{3p-1} T^{p-1} L^{2p}}{\gamma^{p+1}} < \bar{c}_p$, the following results hold:

$$\mathcal{S}_{2p}(\nabla f) \leq C_p \frac{\gamma^{2p-1}}{\eta} \left[\mathbb{E}\|\tilde{V}_{0,0}\|^{2p} + \mathbb{E}|(f(\Psi_0) - f(\Psi_T))^+|^p + 1\right] +$$

$$C_p T \left[\tfrac{\gamma^{4p}}{L^p} + (\gamma\eta T)^{p-1}\gamma^{2p}\right](d + \log k)^p$$

$$\mathcal{S}_{2p}(V) \leq C_p \frac{1}{\gamma\eta} \left[\mathbb{E}\|\tilde{V}_{0,0}\|^{2p} + \mathbb{E}|(f(\Psi_0) - f(\Psi_T))^+|^p + 1\right]$$

$$+ C_p T \left[\tfrac{\gamma^{2p}}{L^p} + (\gamma\eta T)^{p-1}\right](d + \log k)^p$$

**Remark 8.** We believe these bounds are suboptimal. When $V$ is a standard Gaussian random vector, we have $\mathbb{E}\|V\|^{2p} = d^p$. Similarly, when $f$ is L-smooth, it can be shown that

$$\int_{\mathbb{R}^d} \|\nabla f(x)\|^{2p} d\pi(x) \leq \prod_{n=1}^{p} (2n - 1) \cdot (Ld)^p.$$

This is Lemma 12, and is a generalization of Lemma 11 from Vempala & Wibisono (2019). We thus believe the dominant term in both bounds should be $\mathcal{O}(Td^p)$, whereas what we have is $\mathcal{O}(\eta^{p-1}T^p d^p)$. When $T \asymp 1/\alpha\eta$, this is suboptimal in $\kappa$ dependence.

We substitute the bounds from 4, ignoring lower order terms via the assumption $\gamma\eta < c_0$.

$$W_2^2(\text{Law}(\tilde{X}_{t,0}), \text{Law}(X_{t,0})) \lesssim \tfrac{\eta^6 L^4}{\alpha\gamma^2}\left[\mathbb{E}\|\tilde{V}_0\|^2 + \mathbb{E}|(f(\Psi_0) - f(\Psi_T))^+| + 1\right]$$

$$+ \tfrac{\eta^7 L^4}{\alpha\gamma}t\left[\tfrac{\gamma^2}{L} + 1\right](d + \log k) + \tfrac{\eta^6 L^4}{\gamma^4}\left[\mathbb{E}\|\tilde{V}_0\|^4 + \mathbb{E}|(f(\Psi_0) - f(\Psi_T))^+|^2 + 1\right]$$

$$+ \tfrac{\eta^7 L^4}{\gamma^3}t\left[\tfrac{\gamma^4}{L^2} + (\gamma\eta t)\right](d + \log k)^2$$

$$+ \lambda_p \tfrac{\eta^{3p+3} k^p L^{2p+2}}{\gamma^{p+3}}\left[\mathbb{E}\|\tilde{V}_0\|^{2p+2} + \mathbb{E}|(f(\Psi_0) - f(\Psi_T))^+|^{p+1} + 1\right]$$

$$+ \lambda_p \tfrac{\eta^{3p+4} k^p L^{2p+2}}{\gamma^{p+2}}t\left[\tfrac{\gamma^{2p+2}}{L^{p+1}} + (\gamma\eta t)^p\right](d + \log k)^{p+1}$$

$$+ \tfrac{\eta^7 L^4 \gamma}{\alpha^2}(d + \log k) + \tfrac{\eta^8 L^4}{\alpha}(d + \log k)^2 + \lambda_p \tfrac{\eta^{4p+4} k^p L^{2p+2}}{\alpha}(d + \log k)^{p+1}.$$

# G   DEFERRED PROOFS FOR ULMC

## G.1   PROOF OF PROPOSITION 3

*Proof.* Let $\beta = \sqrt{\frac{\eta k}{\gamma}} L \|u_0 - u^+\|$. By Proposition 4 we have $\|G_{\mathcal{M}}(\frac{\eta}{k})^T \Gamma_{\mathcal{M}}(\frac{\eta}{k})^{-2} G_{\mathcal{M}}(\frac{\eta}{k})\| \leq \frac{\eta}{\gamma k}$, and we know $H_{t,i} \leq 1$ since it is a Bernoulli. It follows that $\|S_{t,i}\|^2 \leq \beta^2$. Now let $\nu = \text{Tr}(\mathbb{E}[S_{t,i} S_{t,i}^T | \mathcal{G}])$. Since $\mathbb{E}[(H_{t,i} - 1/k)^2] \leq 1/k$, it follows that $\nu \leq \frac{\eta L^2}{\gamma} \|u_0 - u^+\|^2$. Applying Lemma 1 gives

$$\mathbb{E}[\|Z_{t,i} + S_{t,i} - Y_{t,i}\|^2 | \mathcal{G}] \lesssim \frac{\eta^2 L^4}{\gamma^2} \|u_0 - u^+\|^4 + \mathbf{1}_{5\beta^2 > 1} \cdot \frac{\eta L^2}{\gamma} \|u_0 - u^+\|^2$$

$$\lesssim \frac{\eta^2 L^4}{\gamma^2} \mathbb{E}\|u_0 - u^+\|^4 + \frac{5^p \eta^{p+1} k^p L^{2p+2}}{\gamma^{p+1}} \mathbb{E}\|u_0 - u^+\|^{2p+2}.$$

In the last line, we have used $\mathbf{1}_{5\beta^2 > 1} \leq (5\beta^2)^p = \frac{5^p \eta^p k^p L^{2p}}{\gamma^p} \|u_0 - u^+\|^{2p}$. Multiplying this inequality by $\frac{\eta}{\gamma k}$ finishes the proof. $\qquad\square$

## G.2   PROOF OF LEMMA 10

We will use the following bounds in the proof.

**Lemma 11** (Lemmas 18/19, Kandasamy & Nagaraj (2024)). Let $\Pi : \mathbb{R}^{2d} \to \mathbb{R}^{2d}$ denote projection onto the first $d$ coordinates. Let $G(h)$ and $A(h)$ be as defined in the update rule for underdamped Langevin Monte-Carlo in Section 2.3. Let $\tilde{U}_{t,i}^+, \tilde{V}_{t,i}^+$ and $\tilde{U}_{t,i}, \tilde{V}_{t,i}$ denote the midpoints and iterates respectively of Poisson-ULMC, as defined in Section 2.3. Let $\| \cdot \|$ denote the operator norm of a matrix, and $\| \cdot \|_{\mathbb{R}^n}$ denote the Euclidean norm in dimension $n$. Then the following inequalities are true.

$$\left\| \Pi A_{\mathcal{M}}\left(\frac{j\eta}{k}\right) G_{\mathcal{M}}\left(\frac{\eta}{k}\right) \right\| \lesssim \frac{\eta^2}{k},$$

$$\left\| \begin{bmatrix} \tilde{U}_{t,i}^+ - \tilde{U}_{t,i} \\ \tilde{V}_{t,i}^+ - \tilde{V}_{t,i} \end{bmatrix} \right\|_{\mathbb{R}^{2d}} \leq \sum_{j=0}^{i-1} k H_{t,j} \left\| A\left(\frac{(i-j-1)\eta}{k}\right) G\left(\frac{\eta}{k}\right) \begin{bmatrix} \nabla f(\tilde{U}_{t,0}) - \nabla f(\tilde{U}_{t,i}^+) \\ 0 \end{bmatrix} \right\|_{\mathbb{R}^{2d}}.$$

*Proof.* Recall the definition of $\mathcal{G}$.

$$\mathcal{G} = \left\{ \tilde{X}_{t,0} = y_0 = \begin{bmatrix} u_0 \\ w_0 \end{bmatrix}, \tilde{X}_{t,i} = y = \begin{bmatrix} \tilde{u} \\ \tilde{w} \end{bmatrix}, \tilde{X}_{t,i}^+ = y^+ = \begin{bmatrix} u^+ \\ w^+ \end{bmatrix}, X_{t,i} = x = \begin{bmatrix} u \\ w \end{bmatrix} \right\}.$$

By definition, conditioned on $\mathcal{G}$, we have

$$X_{t,i+1} = A_{\mathcal{M}}\left(\frac{\eta}{k}\right) x + G_{\mathcal{M}}\left(\frac{\eta}{k}\right) \begin{bmatrix} -\nabla f(u) \\ 0 \end{bmatrix} + \Gamma_{\mathcal{M}} Y_{t,i},$$

$$\tilde{X}_{t,i+1} = A_{\mathcal{M}}\left(\frac{\eta}{k}\right) y + G_{\mathcal{M}}\left(\frac{\eta}{k}\right) \begin{bmatrix} -\nabla f(\tilde{u}) \\ 0 \end{bmatrix} + \Gamma_{\mathcal{M}}\left(\frac{\eta}{k}\right) Z_{t,i}$$

$$+ k H_{t,i} \cdot G_{\mathcal{M}}\left(\frac{\eta}{k}\right) \begin{bmatrix} \nabla f(u_0) - \nabla f(u^+) \\ 0 \end{bmatrix}$$

Conditioned on $\mathcal{G}$, we couple $Z_{t,i}, H_{t,i}$ and $Y_{t,i}$ as in Lemma 3. This allows us to define $(X_{t,i+1}, \tilde{X}_{t,i+1})$ using the equations above and gives a conditional coupling of $(Z_{t,i}, H_{t,i}, Y_{t,i}, X_{t,i+1}, \tilde{X}_{t,i+1})$ given $\mathcal{G}$.

We produce an unconditional coupling as follows. Couple $X_{t,i}$ and $\tilde{X}_{t,i}$ optimally w.r.t. $W_2$, then sample $\tilde{X}_{t,i}^+$ and $\tilde{X}_{t,0}$ jointly conditioned on $\tilde{X}_{t,i}$. Conditioned on $(X_{t,i}, \tilde{X}_{t,i}, \tilde{X}_{t,i}^+, \tilde{X}_{t,0})$ we then

sample $(Z_{t,i}, Y_{t,i}, H_{t,i}, X_{t,i+1}, \tilde{X}_{t,i+1})$ from the conditional coupling described above. We now take the expectation in Proposition 2, after substituting the bound in Proposition 3. This gives

$$\mathcal{W}_2^2(X_{t,i+1}, \tilde{X}_{t,i+1}) \le (1 - \Omega(\frac{\alpha\eta}{\gamma k}))\mathcal{W}_2^2(X_{t,i}, \tilde{X}_{t,i}) + E_{t,i}, \text{ where}$$

$$E_{t,i} \lesssim \frac{\eta L^2}{\alpha\gamma k}\mathbb{E}\|\tilde{U}_{t,i}^+ - \tilde{U}_{t,i}\|^2 + \frac{\eta^3 L^4}{\gamma^3 k}\mathbb{E}\|\tilde{U}_{t,i}^+ - \tilde{U}_{t,0}\|^4$$

$$+ \frac{5^p \eta^{p+2} k^{p-1} L^{2p+2}}{\gamma^{p+2}}\mathbb{E}\|\tilde{U}_{t,i}^+ - \tilde{U}_{t,0}\|^{2p+2}.$$

We now bound each of the error terms individually. Recall $N_t := \sum_{i=0}^{k-1} H_{t,i}$ and let $M_{t,k}$ be as defined in Lemma 9.

$$\frac{\eta L^2}{\alpha\gamma k}\mathbb{E}\|\tilde{U}_{t,i}^+ - \tilde{U}_{t,i}\|^2 \le \frac{\eta L^2}{\alpha\gamma k}\mathbb{E}\left[\sum_{j=0}^{i-1} kH_{t,j}\left\|A_{\mathcal{M}}\left(\frac{(i-1-j)\eta}{k}\right)G_{\mathcal{M}}\left(\frac{\eta}{k}\right)\begin{bmatrix}\nabla f(\tilde{U}_{t,0}) - \nabla f(\tilde{U}_{t,i}^+)\\0\end{bmatrix}\right\|\right]^2$$

$$\lesssim \frac{\eta L^2}{\alpha\gamma k}\mathbb{E}\left[\sum_{j=0}^{i-1} H_{t,j} \cdot \eta^2 L\|\tilde{U}_{t,j}^+ - \tilde{U}_{t,0}\|\right]^2$$

$$\lesssim \frac{\eta^5 L^4}{\alpha\gamma k}\mathbb{E}\left[N_t^2 \sup_{0 \le j < k}\|\tilde{U}_{t,j}^+ - \tilde{U}_{t,0}\|^2\right]$$

$$\lesssim \frac{\eta^5 L^4}{\alpha\gamma k}\mathbb{E}\left[\sup_{0 \le j < k}\|\tilde{U}_{t,j}^+ - \tilde{U}_{t,0}\|^2\right]$$

$$\lesssim \frac{\eta^7 L^4}{\alpha\gamma k}\mathbb{E}\|\tilde{V}_{t,0}\|^2 + \frac{\eta^9 L^4}{\alpha\gamma k}\mathbb{E}\|\nabla f(\tilde{U}_{t,0})\|^2 + \frac{\eta^5 L^4}{\alpha\gamma k}\mathbb{E}[M_{t,k}^2]$$

$$\lesssim \frac{\eta^7 L^4}{\alpha\gamma k}\mathbb{E}\|\tilde{V}_{t,0}\|^2 + \frac{\eta^9 L^4}{\alpha\gamma k}\mathbb{E}\|\nabla f(\tilde{U}_{t,0})\|^2 + \frac{\eta^8 L^4}{\alpha k}(d + \log k).$$

In the first inequality, we have used item 2 of Lemma 11. In the second, we have used item 1 of Lemma 11 and Assumption 1. In the fourth we have used that $N_t$ is independent of the iterates, and $\mathbb{E}[N_t]^2 \lesssim 1$. In the fifth and last inequalities, we have used items 1 and 2 of Lemma 9 respectively, with the assumption that $\gamma\eta$ is bounded.

$$\frac{\eta^3 L^4}{\gamma^3 k}\mathbb{E}\|\tilde{U}_{t,i}^+ - \tilde{U}_{t,0}\|^4 \lesssim \frac{\eta^7 L^4}{\gamma^3 k}\mathbb{E}\|\tilde{V}_{t,0}\|^4 + \frac{\eta^{11} L^4}{\gamma^3 k}\mathbb{E}\|\nabla f(\tilde{U}_{t,0})\|^4 + \frac{\eta^3 L^4}{\gamma^3 k}\mathbb{E}[M_{t,k}^4]$$

$$\lesssim \frac{\eta^7 L^4}{\gamma^3 k}\mathbb{E}\|\tilde{V}_{t,0}\|^4 + \frac{\eta^{11} L^4}{\gamma^3 k}\mathbb{E}\|\nabla f(\tilde{U}_{t,0})\|^4 + \frac{\eta^9 L^4}{\gamma k}(d + \log k)^2.$$

The above inequality follows from items 1 and 2 of Lemma 9, with the assumption that $\gamma\eta$ is bounded. Now, for some constant $\lambda_p$ depending only on $p$:

$$\frac{5^p \eta^{p+2} k^{p-1} L^{2p+2}}{\gamma^{p+2}}\mathbb{E}\|\tilde{U}_{t,i}^+ - \tilde{U}_{t,0}\|^{2p+2} \le \lambda_p'\left[\frac{\eta^{3p+4} k^{p-1} L^{2p+2}}{\gamma^{p+2}}\mathbb{E}\|\tilde{V}_{t,0}\|^{2p+2}\right.$$

$$+ \frac{\eta^{5p+6} k^{p-1} L^{2p+2}}{\gamma^{p+2}}\mathbb{E}\|\nabla f(\tilde{U}_{t,0})\|^{2p+2} + \left.\frac{\eta^{p+2} k^{p-1} L^{2p+2}}{\gamma^{p+2}}\mathbb{E}[M_{t,k}^{2p+2}]\right]$$

$$\le \lambda_p\left[\frac{\eta^{3p+4} k^{p-1} L^{2p+2}}{\gamma^{p+2}}\mathbb{E}\|\tilde{V}_{t,0}\|^{2p+2} + \frac{\eta^{5p+6} k^{p-1} L^{2p+2}}{\gamma^{p+2}}\mathbb{E}\|\nabla f(\tilde{U}_{t,0})\|^{2p+2}\right.$$

$$+ \left.\frac{\eta^{4p+5} k^{p-1} L^{2p+2}}{\gamma}(d + \log k)^{p+1}\right].$$

As before, the above inequality follows from items 1 and 2 of Lemma 9. $\qquad\square$

## H    Proof of Corollary 2

*Proof.* Triangle inequality on $\mathcal{W}_2$ gives

$$\mathcal{W}_2^2(\tilde{U}_{N,0}, \pi) \lesssim \mathcal{W}_2^2(\tilde{U}_{N,0}, U_{N,0}) + \mathcal{W}_2^2(U_{N,0}, \pi) \le \mathcal{W}_2^2(\tilde{X}_{N,0}, X_{N,0}) + \mathcal{W}_2^2(U_{N,0}, \pi).$$

Under the conditions of the Corollary, we show that both these terms are $\lesssim \frac{\epsilon^2 d}{\alpha}$. Recall the following Theorem for the convergence of Underdamped LMC.

**Theorem 5** (Corollary of Theorem 2, Dalalyan & Riou-Durand (2020))**.** Let $f$ satisfy Assumption 1. In addition, let the initial condition of ULMC be drawn from the product distribution $\mu = \mathcal{N}(0, \mathbf{I}_d) \otimes \nu_0$. For $\gamma = c\sqrt{L}$ and step-size $h = \frac{0.94\epsilon\sqrt{\alpha}}{L\sqrt{2}}$, the distribution $\nu_k$ of the kth iterate of the ULMC algorithm satisfies $\mathcal{W}_2^2(\nu_k, \pi) \leq \frac{\epsilon^2 d}{\alpha}$ for $k \geq c_3 \frac{\gamma}{\alpha h} \log \frac{\sqrt{2\alpha}W_2(\nu_0, \pi)}{\epsilon\sqrt{d}}$ and some absolute constant $c_3$.

With $k$ defined as in the Corollary we have $\frac{\eta}{k} \lesssim \frac{\epsilon\sqrt{\alpha}}{L}$. Note that the Theorem above is valid with an inequality $h \leq \frac{0.94\epsilon\sqrt{\alpha}}{L\sqrt{2}}$ rather than equality, so we get $\mathcal{W}_2^2(U_{N,0}, \pi) \leq \frac{\epsilon^2 d}{\alpha}$ for $N \geq \frac{c_3\gamma}{\alpha h} \log \frac{\sqrt{2\alpha}W_2(\nu_0, \pi)}{\epsilon\sqrt{d}}$. It remains to be shown that $\mathcal{W}_2^2(\tilde{X}_{N,0}, X_{N,0}) \lesssim \frac{\epsilon^2 d}{\alpha}$. Let $n$ be a natural number. Under our assumptions on $V_{0,0}$ and $U_{0,0}$, we have $\mathbb{E}||V_0||^{2n} = d^n$ and

$$
\begin{aligned}
(f(\Psi_0) - f(\Psi_T))^+ &\leq f(\Psi_0) - f(x^*) \\
&\leq L||\Psi_0 - x^*||^2 \\
&\lesssim L||U_{0,0} - x^*||^2 + \frac{L}{\gamma^2}||V_{0,0}||^2
\end{aligned}
$$

Under our assumptions, we thus get $\mathbb{E}|(f(\Psi_0) - f(\Psi_T))^+|^n \lesssim d^n$. Moreover, we have $\log k \asymp \max(0, \log \frac{\eta L}{\epsilon\sqrt{\alpha}}) \lesssim \log \frac{\kappa}{\epsilon}$ under the assumption that $\gamma\eta < c_0$. Now let $L_2 = c_2 \log \frac{\sqrt{2\alpha}W_2(\nu_0, \pi)}{\epsilon\sqrt{d}}$, $L_3 = \log \frac{\kappa}{\epsilon}$ and apply Theorem 2 with $N$ as above, and

$$
\begin{aligned}
\eta \leq \min \Bigg( &\frac{\epsilon^{1/3}}{\sqrt{L}}, \frac{\epsilon^{1/3}}{\kappa^{1/6}L_2^{1/6}\sqrt{L}}, \frac{\epsilon^{1/3}\kappa^{1/6}}{d^{1/6}\sqrt{L}}, \frac{\epsilon^{1/3}}{\kappa^{1/6}d^{1/6}L_2^{1/3}\sqrt{L}}, \frac{\epsilon^{\frac{p+2}{4p+3}}}{\kappa^{\frac{p/2-1}{4p+3}}d^{\frac{p}{4p+3}}\sqrt{L}}, \\
&\frac{\epsilon^{\frac{p+2}{4p+3}}}{\kappa^{\frac{3p}{8p+6}}d^{\frac{p}{4p+3}}L_2^{\frac{p+1}{4p+3}}\sqrt{L}}, \frac{\epsilon^{1/3}d^{1/6}\kappa^{1/6}}{L_3^{1/6}\sqrt{L}}, \frac{\epsilon^{1/3}d^{1/6}}{\kappa^{1/6}L_3^{1/3}\sqrt{L}}, \frac{\epsilon^{\frac{p+2}{4p+3}}d^{\frac{1}{4p+3}}}{\kappa^{\frac{3p}{8p+6}}L_2^{\frac{p+1}{4p+3}}L_3^{\frac{p+1}{4p+3}}\sqrt{L}} \Bigg).
\end{aligned}
$$

Our assumption on $\epsilon$ is sufficient to ensure that the conditions of Theorem 2 are satisfied with $\eta$ as above. This gives $\mathcal{W}_2^2(\tilde{X}_{N,0}, X_{N,0}) \leq \frac{\epsilon^2 d}{\alpha}$, with $N = L_3\gamma(\alpha\eta)^{-1}$ as desired. $\qquad\square$

# I    TECHNICAL RESULTS FOR ULMC

## I.1    PROOF OF PROPOSITION 2

The following proposition provides useful bounds on the operator norms of $\Gamma_\mathcal{M}$ and $G_\mathcal{M}$ based on Taylor series expansion. We refer to Section I.2 for its proof.

**Proposition 4.** Let $\|\cdot\|$ denote the operator norm of a matrix, and $\|\cdot\|_{\mathbb{R}^n}$ denote the Euclidean norm in dimension $n$. Let $p$ and $q$ denote arbitrary points in $\mathbb{R}^d$. Assume $\gamma h < c_0$ for some sufficiently small constant $c_0 > 0$, and Assumption 1. Then the following inequalities are true.

1. $\|\Gamma_\mathcal{M}(h)\|^2 \lesssim \frac{h}{\gamma}$.

2. $\|G_\mathcal{M}(h)^T \Gamma_\mathcal{M}(h)^{-2} G_\mathcal{M}(h)\| \leq \frac{h}{\gamma}$.

3. $\left\|G_\mathcal{M}(\eta) \begin{bmatrix} \nabla f(p) - \nabla f(q) \\ 0 \end{bmatrix}\right\|_{\mathbb{R}^{2d}} \lesssim \frac{hL}{\gamma}\|p - q\|_{\mathbb{R}^d}$.

We now prove Proposition 2.

*Proof.* Let $h = \frac{\eta}{k}$, and

$$
T\left(\begin{bmatrix} u \\ v \end{bmatrix}\right) = A_\mathcal{M}(h)\begin{bmatrix} u \\ v \end{bmatrix} + G_\mathcal{M}(h)\begin{bmatrix} -\nabla f(u) \\ 0 \end{bmatrix}.
$$

Given Assumption 1, with $\gamma = c\sqrt{L}$ for some $c \geq \sqrt{2}$, the map $T$ is Lipschitz with $\|T\|_{\text{Lip}} \leq 1 - \frac{\alpha}{\sqrt{L}}h + O(Lh^2)$ (Lemma 16, Zhang et al. (2023).) Under our assumptions we have $L(\frac{\eta}{k})^2 \lesssim \frac{\alpha}{\sqrt{L}} \cdot \frac{\eta}{k}$, and $T$ is thus a contraction with parameter $1 - \Omega(\frac{\alpha\eta}{\sqrt{L}k})$. Under the event $\mathcal{G}$, we have

$$\left\|X_{t,i+1} - \tilde{X}_{t,i+1}\right\|^2 = \left\|T(x) - T(y)\right\|^2 + \left\|\Gamma_{\mathcal{M}}\left(\frac{\eta}{k}\right)(Y_{t,i} - \tilde{Z}_{t,i})\right\|^2$$
$$+ 2\left\langle\Gamma_{\mathcal{M}}\left(\frac{\eta}{k}\right)(Y_{t,i} - \tilde{Z}_{t,i}), T(x) - T(y)\right\rangle.$$

By the definition of $\tilde{Z}_{t,i}$,

$$\Gamma_{\mathcal{M}}\left(\frac{\eta}{k}\right)(\tilde{Z}_{t,i} - Y_{t,i}) = \Gamma_{\mathcal{M}}\left(\frac{\eta}{k}\right)(Z_{t,i} + S_{t,i} - Y_{t,i}) + G_{\mathcal{M}}\left(\frac{\eta}{k}\right)\begin{bmatrix}\nabla f(\tilde{U}_{t,i}) - \nabla f(\tilde{U}_{t,i}^+) \\ 0\end{bmatrix}.$$

Conditioned on $\mathcal{G}$, $(H_{t,i} - 1/k)$ is zero mean and $Z_{t,i}$ and $Y_{t,i}$ are standard Gaussians. This gives $\mathbb{E}[\Gamma_{\mathcal{M}}(\frac{\eta}{k})(Z'_{t,i} - \tilde{Z}_{t,i})|\mathcal{G}] = G_{\mathcal{M}}\left(\frac{\eta}{k}\right)\begin{bmatrix}\nabla f(\tilde{U}_{t,i}) - \nabla f(\tilde{U}_{t,i}^+) \\ 0\end{bmatrix}$. By item 3 of Proposition 4, and the contractivity of $T$, we get

$$\mathbb{E}[\|X_{t,i+1} - \tilde{X}_{t,i+1}\|^2|\mathcal{G}] \leq (1 - \Omega(\frac{\alpha\eta}{\gamma k}))\|x - y\|^2 + \mathcal{O}\Big[\left\|\Gamma_{\mathcal{M}}\left(\frac{\eta}{k}\right)(Y_{t,i} - \tilde{Z}_{t,i})\right\|^2$$
$$+ \frac{\eta L}{k\gamma}\|u^+ - \tilde{u}\| \cdot \|x - y\|\Big].$$

An application of the AM-GM inequality gives

$$\frac{\eta L}{\gamma k}\|u^+ - \tilde{u}\| \cdot \|x - y\| \lesssim \frac{\eta L^2}{\tau\alpha\gamma k}\|u^+ - \tilde{u}\|^2 + \tau\frac{\alpha\eta}{\gamma k}\|x - y\|^2,$$

Where $\tau > 0$ is arbitrary. Choose $\tau$ small enough so that the second term can be absorbed into $(1 - \Omega(\frac{\alpha\eta}{\gamma k}))\|x - y\|^2$. We also have

$$\mathbb{E}[\|\Gamma_{\mathcal{M}}\left(\frac{\eta}{k}\right)(\tilde{Z}_{t,i} - Y_{t,i})\|^2|\mathcal{G}] \lesssim \|G_{\mathcal{M}}\left(\frac{\eta}{k}\right)\begin{bmatrix}\nabla f(\tilde{U}_{t,i}) - \nabla f(\tilde{U}_{t,i}^+) \\ 0\end{bmatrix}\|^2$$
$$+ \|\Gamma_{\mathcal{M}}\left(\frac{\eta}{k}\right)\|^2 \cdot \mathbb{E}[\|Z_{t,i} + S_{t,i} - Y_{t,i}\|^2|\mathcal{G}].$$

By item 3 of Proposition 4, $\|G_{\mathcal{M}}\left(\frac{\eta}{k}\right)\begin{bmatrix}\nabla f(\tilde{U}_{t,i}) - \nabla f(\tilde{U}_{t,i}^+) \\ 0\end{bmatrix}\|^2 \leq \frac{\eta^2 L^2}{k^2\gamma^2}\|\tilde{u} - u^+\|^2 \leq \frac{\eta L^2}{\alpha\gamma k}\|\tilde{u} - u^+\|^2$; and by item 1, $\|\Gamma_{\mathcal{M}}\left(\frac{\eta}{k}\right)\|^2 \leq \frac{\eta}{\gamma k}$.

$$\mathbb{E}[\|\Gamma_{\mathcal{M}}(\frac{\eta}{k})(\tilde{Z}_{t,i} - Y_{t,i})\|^2|\mathcal{G}] \lesssim \frac{\eta L^2}{\alpha\gamma k}\|\tilde{u} - u^+\|^2 + \frac{\eta}{\gamma k}\mathbb{E}[\|Z_{t,i} + S_{t,i} - Y_{t,i}\|^2|\mathcal{G}].$$

$\square$

### I.2 PROOF OF PROPOSITION 4

*Proof.* The eigenvalues of $\Gamma_{\mathcal{M}}(h)^2$ are

$$E_1 = \frac{\exp(-2\gamma h)}{\gamma^2}(a - b), \qquad\qquad E_2 = \frac{\exp(-2\gamma h)}{\gamma^2}(a + b)$$

where $a = -1 + \exp(2\gamma h)(1 + 2\gamma h)$, and

$$b = \sqrt{1 - 32\exp(3\gamma h) + 2\exp(2\gamma h)(7 + 2\gamma h) + \exp(4\gamma h)(17 - 4\gamma h + 4\gamma^2 h^2)}.$$

Taylor expansion in the variable $h$ gives

$$\frac{\exp(-2\gamma h)}{\gamma^2}a = \frac{4h}{\gamma} - 2h^2 + \frac{4\gamma h^3}{3} + O(\gamma^2 h^4),$$

$$\frac{\exp(-2\gamma h)}{\gamma^2}b = \frac{4h}{\gamma} - 2h^2 + \frac{7\gamma h^3}{6} + O(\gamma^2 h^4).$$

As a result, the eigenvalues $E_1$ and $E_2$ are of order $\gamma h^3$ and $\frac{h}{\gamma}$ respectively, with $E_2 \geq E_1$ being the spectral norm of $\Gamma_{\mathcal{M}}(h)^2$. We compute the inverse:

$$\det(\Gamma_{\mathcal{M}}(h)^2) = \frac{8}{\gamma^4}\exp(-2\gamma h)(-1 + \exp(\gamma h))(2 + \gamma h + \exp(\gamma h)(-2 + \gamma h)),$$

$$\Gamma_{\mathcal{M}}(h)^{-2} = \det(\Gamma_{\mathcal{M}}(h)^2)^{-1}\begin{bmatrix} \frac{4(1-\exp(-\gamma h))-(1-\exp(2\gamma h))+2\gamma h}{\gamma^2}\mathbf{I}_d & -\frac{2\gamma h - (1-\exp(2\gamma h))}{\gamma^2}\mathbf{I}_d \\ -\frac{2\gamma h - (1-\exp(2\gamma h))}{\gamma^2}\mathbf{I}_d & \frac{4(1-\exp(-\gamma h))+(1-\exp(2\gamma h))+2\gamma h}{\gamma^2}\mathbf{I}_d \end{bmatrix}.$$

An explicit computation gives

$$G_{\mathcal{M}}(h)^T\Gamma_{\mathcal{M}}(h)^{-2}G_{\mathcal{M}}(h) = \begin{bmatrix} \frac{h}{2\gamma} & 0 \\ 0 & 0 \end{bmatrix}.$$

A Taylor expansion on the entries of $G_{\mathcal{M}}(h)$ shows

$$G_{\mathcal{M}}(h) = \begin{bmatrix} \frac{h^2}{2} + O(\gamma h^3) & 0 \\ \frac{2h}{\gamma} - \frac{h^2}{2} + O(\gamma h^3) & 0 \end{bmatrix}.$$

Item 3 of the proposition follows from this and the smoothness of $f$ – Assumption 1. $\qquad\square$

**Lemma 12.** Assume $\pi = \exp(-f)$ is $L$-smooth, and let $p \in \mathbb{N}$. Then

$$\int_{\mathbb{R}^d} ||\nabla f(x)||^{2p}d\pi(x) \leq \prod_{n=1}^{p}(2n-1)\cdot(Ld)^p.$$

*Proof.* This is a generalization of Vempala & Wibisono (2019, Lemma 11). By definition, we have

$$\int_{\mathbb{R}^d} ||\nabla f(x)||^{2p}d\pi(x) = \int_{\mathbb{R}^d}\exp(-f(x))\Big[\sum_{i=1}^d\Big(\frac{\partial f}{\partial x_i}\Big)^2\Big]^p dx$$

By Jensen's inequality, we get

$$\leq d^{p-1}\sum_{i=1}^d\int_{\mathbb{R}^d}\exp(-f(x))\Big(\frac{\partial f}{\partial x_i}\Big)^{2p}dx$$

Applying integration by parts along $x_i$, we get

$$= d^{p-1}(2p-1)\sum_{i=1}^d\int_{\mathbb{R}^d}\exp(-f(x))\Big(\frac{\partial^2 f}{\partial x_i^2}\Big)\Big(\frac{\partial f}{\partial x_i}\Big)^{2p-2}dx$$

Since $f$ is $L$-smooth, we have $\frac{\partial^2 f}{\partial x_i^2} \leq L$.

$$\leq Ld^{p-1}(2p-1)\sum_{i=1}^d\int_{\mathbb{R}^d}\exp(-f(x))\Big(\frac{\partial f}{\partial x_i}\Big)^{2p-2}dx$$

By a repeated application of integration by parts, we get

$$\leq L^p d^{p-1}\prod_{n=1}^p(2n-1)\sum_{i=1}^d\int_{\mathbb{R}^d}\exp(-f)dx$$

Since $\pi$ is a probability measure, $\int\exp(-f) = 1$.

$$= \prod_{n=1}^p(2n-1)(Ld)^p.$$

$$\square$$

