# OpenReview forum: "Poisson Midpoint Method for Log Concave Sampling: Beyond the Strong Error Lower Bounds"
_ICLR.cc/2026/Conference — ICLR 2026 Poster_

### Official Review · Reviewer_HCmB · 2025-10-30

**Soundness:** 3
**Presentation:** 2
**Contribution:** 4
**Rating:** 6
**Confidence:** 4

**Summary:**

This paper studies the Wasserstein-2 convergence of both the overdamped and underdamped Poisson Randomized Midpoint Methods (PLMC) for sampling from *strongly log-concave* distributions, with a focus on the dependencies of dimension ($d$), accuracy ($\varepsilon$) and condition number ($\kappa$) in the oracle complexity.

The analysis builds on the intuition that one step PLMC of size $\eta$ *approximately implement* $k$ step of LMC with size $\eta/k$. Accordingly, the convergence proof of PLMC is devided into two parts: convergence analysis of LMC and **trajectory-gap quantification between LMC and PLMC**.  The main theoretical novelty lies in the latter, where the authors construct a refined coupling between the Brownian motions driving the two dynamics, enabling a tight control of their deviation. Combining this with existing convergence results for overdamped LMC ([1]) and underdamped LMC ([2]), sharper bounds in terms of the accuracy dependency are proved: $\mathcal{O}(\varepsilon^{-2/3})$ for overdamped PLMC and $\mathcal{O}(\varepsilon^{-1/3})$ for underdamped PLMC.

[1] Alain Durmus, Szymon Majewski, and Bła˙ zej Miasojedow. Analysis of Langevin Monte Carlo via
Convex Optimization. The Journal of Machine Learning Research, 20(1):2666–2711, 2019.

[2] Arnak Dalalyan and Lionel Riou-Durand. On sampling from a log-concave density using kinetic
Langevin diffusions. Bernoulli, 26(3), 2020.

**Strengths:**

1. The theoretical derivation in the paper is solid.

2. The complexity bound beats all existing bounds in terms of accuracy dependency.

3. The improved bounds also help to understand an misunderstanding on the information lower bound derived in [1], suggesting that better bounds than $\mathcal{O}(\varepsilon^{-2/3})$ can be achieved if the desired accuracy metric is $W_2$/ KL-divergence.

**Weaknesses:**

The presentation of the paper require improvement. As far as I am concerned, more discussion on

(1) good performance of PLMC compared to LMC;

(2) intuition behind of the constructed coupling, along with why it implies better complexity;

(3) whether the current derived bound is sharp;

should be added.

**Questions:**

1. Typos:

(1) line 85-86: at the end of the line, *for underdamped LMC* should be *for underdamped PLMC*;

(2) line 194-197: the summation should be from $j=0$ to $j=i-1$.

2. The proofs can be organized in a better way. For example, both the high-level idea for convergences of overdamped and underdamped PLMC should be talked about. Currently, the part related to underdamped PLMC is not very informative.

---

> ### Author Response · Authors · 2025-11-19
> **Author Response.**
>
> We thank the reviewer for the kind review, noting the significance of our contributions, and for pointing out typographical errors, which we have fixed.  Here are some clarifying remarks corresponding to the points raised by the reviewer. We will add these to the manuscript.
>
> **Why PLMC is better than LMC:** The improvement observed is due to PLMC being a better approximator of the underlying stochastic process (Langevin dynamics). The key to our improvement is Lemma 1, which (roughly) states the following:
>
> Let $V$ be a random vector with zero mean and variance $\mathbb{E}||V||^2 = \nu$. If $Z$ is an independent standard Gaussian, then
>
> $$W_2^2(\mathsf{Law}(Z), \mathsf{Law}(Z+V)) \lesssim \nu^2 \,.$$
>
> A straight forward coupling, often used in prior works, shows $W_2^2 \leq \nu$. This could potentially be much larger than our bound. To intuitively explain this bound and show that our derivation is sharp, consider the scenario where f $V \sim \mathcal{N}(0,\nu) \in \mathbb R$ and $Z \sim \mathcal{N}(0,1)$ . Then, there is a closed form expression for $W_2^2(\mathsf{Law}(Z), \mathsf{Law}(Z+V))$:
>
> $$W_2^2(\mathsf{Law(Z)}, \mathsf{Law(Z+V)}) = 2+\nu - 2\sqrt{1+\nu}  = \Theta(\nu^2). \text{(when $\nu \ll 1$)  (Lines 628-631)}$$
>
>
> **Sharpness of current bound:** We believe that the coupling shown above is sharp due to the Gaussian case shown. We have added this comment to the manuscript (Lines 416-421). However, when the entire algorithm is considered, lower bounds in this setting are not known. As explained in the paper, computational complexity lower bounds for sampling algorithms is still largely an open problem with very few results found in the literature. Unlike statistical learning theory, there are no systematic methods to prove such bounds. Thus, we believe this is beyond the scope of our present work.
>
> However, we believe our dependence on $\kappa$ in Corollary 2 can be improved if we obtain tight bounds on the higher order moments of our algorithm. This is explained in Section 5 and Remark 8, and in our revision we have noted this below Corollary 2 (Lines 394-395). We hope to explore this in future work.

---

> > ### Comment · Reviewer_HCmB · 2025-11-26
> >
> > I appreciate the authors’ clarifications. My questions have been fully addressed, and the authors have partially addressed the weaknesses I raised. However, regarding points (2) and (3), I believe a more comprehensive discussion of the coupling techniques used in analyzing underdamped/overdamped PLMC and RLMC is essential, but this remains absent in the current manuscript. Therefore, I will maintain my original evaluation.

---

### Official Review · Reviewer_SpWw · 2025-10-31

**Soundness:** 3
**Presentation:** 3
**Contribution:** 3
**Rating:** 6
**Confidence:** 5

**Summary:**

This paper studies sampling from strongly log-concave distributions. It establishes that the Poisson midpoint discretization of both overdamped and underdamped Langevin dynamics achieves an accelerated convergence rate under the  W2 distance, significantly improving upon existing bounds.

**Strengths:**

he paper is very well-written. It provides a clear description of the problem, a thorough comparison of its theoretical results with existing works, and insightful proof techniques that illustrate the reasons behind the achieved convergence rates.

**Weaknesses:**

It is recommended to exchange the positions of Sections 2.2 and 2.3 for a more logical flow. Additionally, the origin of the Poisson midpoint method should be clarified: is it a novel proposal of this work, or is it adopted from prior literature?

**Questions:**

1. To better highlight the novelty of the convergence rate, could the authors provide a more detailed explanation of the limitations in prior work that prevented them from achieving the same rate?

2. The role of the parameter p in the analysis should be clarified. Furthermore, does the method maintain its performance advantage for ill-conditioned problems where the condition number kapa is very large?

3. Could the authors include numerical experiments to empirically validate the theoretical findings?

---

> ### Author Response · Authors · 2025-11-19
> **Author Response.**
>
> We thank the reviewer for the kind review and the suggestions for improving our work. We have swapped sections 2.2 and 2.3 as recommended.
>
> **Detailed explanation of limitations of prior works:** In all prior works studying convergence in $W_2$, the upper bounds are derived by a loose coupling where the Brownian motions driving the algorithm and the continuous time process are coupled identically. This “strong L2” coupling is sub-optimal, as we show in our work. Indeed, the work of [Cao et. al.] shows that the rate of $\epsilon^{-2/3}$ cannot be beaten by such arguments. As a result, this rate hasn’t been improved since it was originally established in 2019 [Shen & Lee]. Our primary technical breakthrough is to produce a superior coupling via the Wasserstein CLT due to Zhai (Lemma 1.) This allows us to breach the barrier established by Cao. et. al. and obtain a novel convergence rate.
>
> **The parameter $p$:** shows up because we use a bound of the form $1(\beta > 1) \leq \beta^p$. Taking expectation on both sides leads to the p-th moment. This bound holds for any $p$, but we can choose various values of p to achieve different trade offs. This is noted in lines 364-366, and Remark 7. We are happy to provide further clarification.
>
> **Ill-conditioned log-concave sampling:** In the ill-conditioned regime, there are no results to our knowledge for convergence in the Wasserstein distance. It is more common to see guarantees in KL divergence [Durmus et. al. (2019), Altschuler et. al. (2025), Altschuler & Chewi (2024)].  In this case, the coupling argument from our work cannot be used to show convergence.  We believe shifted composition based methods in [Altschuler et. al. (2025), Altschuler & Chewi (2025)] can be used here. However, this method only achieves $1/\epsilon^{2/3}$ complexity and additional research is required to improve this rate.
>
>
> **Regarding experiments:** Prior works on randomized midpoint methods have considered a diverse range of experiments and have shown its efficacy for strongly log-concave sampling [Shen & Lee (2019)] and even for generative modeling with DDPMs [Kandasamy & Nagaraj (2024)]. Thus we chose to focus on theoretical guarantees in our work as done in most papers in this area [Cao et. al (2021), He et. al (2021), Yu et. al (2024), Altschuler & Chewi (2024), Altschuler et. al (2025)].

---

### Official Review · Reviewer_k3Ji · 2025-11-01

**Soundness:** 3
**Presentation:** 2
**Contribution:** 4
**Rating:** 8
**Confidence:** 4

**Summary:**

The paper analyzes the Poisson midpoint method (proposed in an earlier work) in the strongly convex setting. It obtains rates which scale as $\varepsilon^{-1/3}$ with respect to the desired inverse accuracy, which bypasses conjectured lower bounds. It does so using a Wasserstein central limit theorem.

**Strengths:**

The paper obtains surprising low accuracy sampling guarantees with oracle complexity scaling as $\varepsilon^{-1/3}$. This defeats a conceptual lower bound.

The paper obtains these new rates in a conceptually orthogonal way to any prior analysis, which potentially opens the doors to successive works adapting these techniques to other settings.

I do not want to overelaborate, but I think these are clear and very salient contributions, and I believe the techniques in this paper deserve more exposure.

**Weaknesses:**

In principle, improved rates for low accuracy algorithms in $\varepsilon$ are not useful if they come at the expense of dimension dependence, due to the existence of high accuracy algorithms. In particular, if we take $\varepsilon = \tilde{\varepsilon}/d^{1/2}$ here, then we observe worse dimension dependence $d^{3/8}$ in the second term (as $p \to \infty$). However, I do not see this as a severe issue.

This paper explores only the simplest principled setting; many settings still fall outside the purview of this paper, for instance the LSI setting or the low friction setting.

**Questions:**

The comparison with Altschuler et al. (2025) is made twice, which seems redundant.

Can this be adapted to SDEs without convergent drifts? This may be helpful to numerical analysts.

38: Ito -> It\^o

41-42 is not a complete sentence.

The guarantee $W_2^2 \lesssim \frac{\varepsilon^2 d}{\alpha}$ is non-standard in terms of the quoted rates. It would be preferable if one absorbed this $d$ into the factor $\varepsilon^2$, which is always possible. Indeed, this is done in Table 1 and it is a bit puzzling why this is not done elsewhere.

---

> ### Author Response · Authors · 2025-11-19
> **Author Response**
>
> We thank the reviewer for the kind review, recognizing the importance of our contributions; and for pointing out typographical errors, which we have rectified.
>
> **Dimension Dependence:** While our dimension dependence is worse, as pointed out by the reviewer, this is not a severe issue since our primary purpose is to probe the fundamental limits of various classes of sampling algorithms. We also hope that our methods will be insightful towards future work.
>
> **Time Dependent Drifts:** This method can be extended to SDEs with time dependent drifts as shown in [Kandasamy and Nagaraj (2024)]. However, their work only considered empirical evaluation for denoising diffusion models. Post our submission, we saw the following work which also provides theoretical guarantees: arxiv 2511.04844 . We hope to consider analyzing this method and its efficacy in the numerical analysis context in future work.
>
> **Convention for Bounds:** The convention $W_2^2 ≤ \epsilon^2d/\alpha$ is adopted from the original work on the randomized midpoint method [Shen & Lee (2019)] and has since become standard in the randomized midpoint literature. As the reviewer has noted, another common convention is $W_2^2 ≤ \epsilon^2/\alpha$, which we have also considered.
>
> We made the present choice of notation since the table compares against other metrics such as KL and TV, making the latter convention useful,  whereas the theorems are more likely to be compared to prior works on randomized midpoint methods with $W_2$ convergence, where the former can be helpful. **If the reviewer still believes it would be helpful to state results uniformly, we will make the change.**

---

### Official Review · Reviewer_Pskp · 2025-11-01

**Soundness:** 4
**Presentation:** 3
**Contribution:** 3
**Rating:** 6
**Confidence:** 2

**Summary:**

The paper studies the Poisson Midpoint Method (PLMC) for overdamped and underdamped Langevin dynamics on strongly log‑concave targets. It proves Wasserstein‑2 convergence with improved oracle complexities: for overdamped (Cor. 1), and for underdamped (Cor. 2),. The analysis combines a tight W2 bound for Gaussian + 1-D perturbation (Lemma 1, adapted from Zhai) with contractive couplings. Tables 1–2 (p. 6) compare against LMC/RLMC and emphasize a cubic speedup in (ε) over Euler-Maruyama. The paper also clarifies the distinction between strong (L2) error lower bounds and weak W2 guarantees.

**Strengths:**

Important question, clear positioning: The introduction and §1.1 make a precise case that strong (L2) lower bounds for ULD do not preclude faster W2 rates, and the results indeed obtain $\tilde O(ε^{-1/3})$ for the underdamped case (Theorem 2, Cor. 2).

Clear presentation on technical novelty and algorithmic efficiency.

**Weaknesses:**

Typo: In Eq (2), the coefficient on Brownian term should be $\sqrt{2\gamma}d$, in order to achieve right invariant distribution.

No empirical study: The paper is purely theoretical. There are no experiments illustrating constants, stability, or the practical effect of hyper-parameters.

**Questions:**

No.

---

> ### Author Response · Authors · 2025-11-19
> **Author Response.**
>
> We thank the reviewer for acknowledging the impact of our theoretical contributions and the clear presentation of our ideas, and pointing out the typographical error in equation (2). We believe they meant $\sqrt{2\gamma}dB_t$, and have corrected this in the manuscript.
>
> **Regarding experiments:** Prior works on randomized midpoint methods have considered a diverse range of experiments and have shown its efficacy for strongly log-concave sampling [Shen & Lee (2019)] and even for generative modeling with DDPMs [Kandasamy & Nagaraj (2024)]. Thus we chose to focus on theoretical guarantees in our work as done in most papers in this area [Cao et. al (2021), He et. al (2021), Yu et. al (2024), Altschuler & Chewi (2024), Altschuler et. al (2025)].

---

### Meta-Review · Area_Chair_6RA9 · 2025-12-25

**Summary:**

Essentially, no major concerns were raised. Some minor concerns:
- Lack of experimental validation. This does not seem to be a notable detriment, as the focus is purely theoretical.
- Results are only presented for the strongly log-concave setting. This seems inherent to the proof technique which works in $W_2$, but extending beyond this setting seems to be out of scope.
- Some issues were raised regarding the exposition (e.g., providing intuition about the source of improvements, and maintaining consistency with regards to the target accuracy throughout). The authors should take these issues seriously and revise the draft for the sake of readability.

Overall, this is a solid theoretical contribution and clearly meets the bar for acceptance.

**Reviewer Concerns:**

See above.

**Reviewer Scores:**

The reviewers would have kept the scores unchanged, more or less.

---

### Decision · Program_Chairs · 2026-01-26

Accept (Poster)